# Understanding Long Videos with Multimodal Language Models

**Kanchana Ranasinghe, Xiang Li, Kumara Kahatapitiya & Michael S. Ryoo**
`kranasinghe@cs.stonybrook.edu`

## Abstract

Large Language Models (LLMs) have allowed recent LLM-based approaches to achieve excellent performance on long-video understanding benchmarks. We investigate how extensive *world knowledge* and strong *reasoning skills* of underlying LLMs influence this strong performance. Surprisingly, we discover that LLM-based approaches can yield surprisingly good accuracy on long-video tasks with limited video information, sometimes even with no *video-specific* information. Building on this, we explore injecting video-specific information into an LLM-based framework. We utilize off-the-shelf vision tools to extract three object-centric information modalities from videos, and then leverage natural language as a medium for fusing this information. Our resulting Multimodal Video Understanding (MVU) framework demonstrates state-of-the-art performance across multiple video understanding benchmarks. Strong performance also on robotics domain tasks establishes its strong generality. Code: github.com/kahnchana/mvu

## 1 Introduction

> What can we learn from videos,
> beyond scene context understood from a single natural image?

Recent success of large language models (LLMs) and their visual extensions, vision-language models (VLMs), has led to incredible performance on complex language-tied video understanding benchmarks (Zhang et al., 2023a), particularly on long-video question answering: a task that requires awareness over longer temporal windows (Mangalam et al., 2023) as well as causal and temporal action reasoning (Xiao et al., 2021). However, the LLMs underlying these approaches contain extensive world knowledge (e.g. understanding of physics, culture, human common sense) and reasoning abilities (Yu et al., 2023a; Wang & Zhao, 2023), raising the question of whether they excel at video tasks due to actual *video modality* awareness or simply utilizing world knowledge and contextual information. Such understanding of model reasoning is important for robust deployments avoiding spurious correlation based predictions as well as for better model interpretability (Yun et al., 2022; Xiao et al., 2024).

In this work, we systematically study this question in the context of video question-answering (QnA) benchmarks, building two modality-constrained baselines to highlight our findings. These two frameworks are tagged *Just-LLM* and *Single-Frame-VLM*. The first is constrained to access only the task textual query (i.e. no task-specific visual information). The latter is given access to task context with an additional single center-frame from the video as input. We discover how these models perform significantly better than random prediction on multiple long-video understanding benchmarks (see Table 1, similar findings in Min et al. (2024)). In fact, the latter, utilizing purely world knowledge and contextual information, even outperforms multiple recent state-of-the-art video understanding works (see Table 2), challenging the notion of how much *video information* is actually utilized by existing approaches to solve these complex video QnA tasks.

We next focus on efficient inference to allow rapid experimentation with our LLM based frameworks. Therein, we explore suitable prompting and templating to adapt likelihood selection techniques from prior work (Robinson et al., 2023) to video QnA tasks. Our resulting framework achieves more efficient inference with improved performance in comparison to prior work that com-

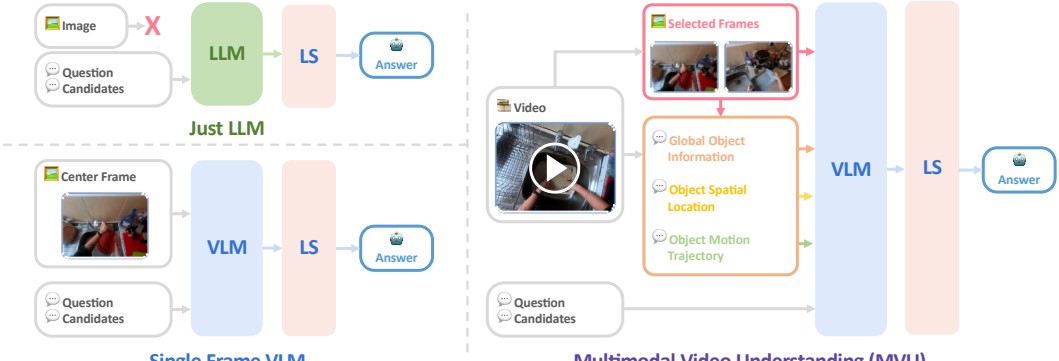

Figure 1: **Overview of Framework:** We propose three variants of our framework that solves complex long-video question-answering tasks. (left-top) Just-LLM utilizes only world knowledge with zero task-specific awareness. (left-bottom) Single-Frame-VLM processes an additional center frame to obtain task context but accesses no *video* specific information. (right) Our complete approach, MVU extracts three additional object-centric information modalities followed by fusion in language space. LS refers to likelihood selection.

monly use auto-regressive generation to tackle long-video QnA benchmarks (Zhang et al., 2023a; Balavzevi'c et al., 2024; Wang et al., 2025).

Motivated by our initial findings on modality-constrained performance, we study how to inject additional video-specific information into our framework using natural language in a concise and interpretable manner to further improve video understanding. We explore three forms of *object-centric* information modalities, develop pipelines requiring zero video-level training to extract such information using off-the-shelf vision tools, and utilize natural language to fuse this multi-modal information using templating operations. Our resulting approach, termed Multi-Modal Video Understanding (MVU) framework, while achieving state-of-the-art zero-shot performance across long-video understanding benchmarks, also exhibits better interpretability (e.g. exposing video-specific information utilized) through its language-based operation. Moreover, our proposed MVU exhibits generality with its strong performance even on robotics domain tasks.

In summary, our key contributions are as follows:

1. Uncover surprisingly strong performance on complex video-language tasks by modality-constrained baselines with limited access to video-specific information.
2. Adapting Likelihood Selection strategies to video QnA benchmarks for efficient evaluation.
3. Novel VLM-based video QnA framework that extracts concise video specific object-centric information followed by natural language based fusion.

We integrate our MVU framework over multiple different baselines and obtain performance improvements across 20 different datasets establishing both its effectiveness and generality. Our evaluations are performed zero-shot with no video-level training on these datasets which cover video QnA tasks (short, medium, and long videos) as well robotics domain tasks.

## 2 RELATED WORK

**Video Modality Exploration:** Multiple recent works dissect the video modality into individual components (Yun et al., 2022; Buch et al., 2022; Ranasinghe et al., 2021; Ramasinghe et al., 2018). Single frame baselines are one interesting sub-class (Buch et al., 2022; Davis & Bobick, 1997; Zhao et al., 2017; Safaei & Foroosh, 2019; Bilen et al., 2016). Extracting object-centric video modalities is another idea, spanning back to Davis & Bobick (1997) which extracts multiple small objects from frames followed by modeling relations across frames and objects. Similarly, Safaei & Foroosh (2019); Zhao et al. (2017) combine spatial information with single images to perform video tasks. However, these prior approaches focus on simple video tasks (i.e. action recognition) limited to visual modality. In contrast, our approach tackles the more complex language-tied task of long-video question answering that necessitates strong causal and temporal reasoning over long temporal windows. This task is also explored in Buch et al. (2022), but we differ with likelihood selection, multi-modal information fusion, and usage of modern LLMs.

**Long Video Question Answering:** Long-video question-answering benchmarks are constructed to specifically test strong causal and temporal reasoning (Xiao et al., 2021) over long temporal windows (Mangalam et al., 2023). Early works explore querying objects or events based on referential and spatial relations (Xu et al., 2017; Zeng et al., 2017; Yu et al., 2019), followed by focus on temporal modeling of sequential events (Lei et al., 2018; 2020; Hosseini et al., 2022; Xiao et al., 2022a;b). While motivated by these works, MVU integrates such object information with large language models (LLMs) in a zero-shot manner requiring no video-level training. More recent works leverage LLMs (Yu et al., 2023b; Papalampidi et al., 2023; Wang et al., 2024b; Balavzevi'c et al., 2024; Wang et al., 2024a) to directly perform these tasks but require video-caption training. In contrast, our MVU operates zero-shot on these tasks requiring no video-level training. Zero-shot operation is explored in Wang et al. (2023a); Zhang et al. (2023a); Min et al. (2024); Wang et al. (2023b; 2024c), but we differ in using object-centric information modalities and efficient LLM sampling.

**Large Language Model Reasoning:** Recent LLMs (OpenAI, 2023; Chowdhery et al., 2022; Chiang et al., 2023) demonstrate multiple forms of strong reasoning abilities (Kıcıman et al., 2023; Creswell & Shanahan, 2022; Liu et al., 2023b) including combining different information (Weston & Sukhbaatar, 2023). Their recent open-source variants (Touvron et al., 2023; Team et al., 2023; Jiang et al., 2023) achieve equally promising skills using scaled-down models (Jiang et al., 2023) while also demonstrating strong world knowledge (Yu et al., 2023a; AlKhamissi et al., 2024; Zhao et al., 2023; Wang & Zhao, 2023; Xu et al., 2024; Li et al., 2023d) even in domains such as robotics (Li et al., 2024). In our work, we leverage these strengths of LLMs for complex video-language tasks, focused on disentangling the effect of their abilities for video QnA tasks.

**Language based Fusion:** The idea of fusing different modality information using natural language as a medium has been explored in multiple recent works (Ranasinghe & Ryoo, 2023; Lin et al., 2023b; Hanu et al., 2022; Wang & Chen, 2017; Hanu et al., 2023; Zeng et al., 2022). In Ranasinghe & Ryoo (2023); Lin et al. (2023b), language is utilized as an implicit medium for self-supervising video action recognition. Multimodal information represented as language is fused with visual information for action recognition and robotics tasks in (Hanu et al., 2022; Wang & Chen, 2017; Hanu et al., 2023; Li et al., 2024). We utilize a similar language-as-a-medium fusion of multimodal information, but explore this in the context of complex video-language tasks. Zeng et al. (2022) is most similar to our work, but we differ with focus on long-video tasks and object-centric information.

## 3 NAIVE BASELINES & LIKELIHOOD SELECTION

In this section, we first establish our problem setting, then discuss adapting likelihood selection for video QnA tasks, and finally introduce two naive LLM based frameworks for video question answering tasks, tagged *Just-LLM* and *Single-Frame-VLM* (see Figure 1).

### 3.1 PROBLEM FORMULATION

We focus on two categories of video understanding tasks:

1. Long Video Question Answering (Multiple-Choice-based Selection)
2. Open Ended Video Question Answering (Text Generation)

For the first task, we construct a unified problem formulation accounting their choice based selection aspect. For the latter, we resort to standard LLM based answer generation.

Consider a video $x_v \in \mathbb{R}^{L \times H \times W \times C}$, a textual question $x_t$, a set of textual candidate answers $Y = \{y_i, i = 1, ..., M\}$, and a model $V(\cdot)$ selecting one answer from the given set of answers (noted as $\hat{y} := V(x_v, x_t, Y)$). Selected $\hat{y}$ should ideally be identical to groundtruth $y_g$. Here $L, H, W, C$ are the number of frames of the video, frame height, width, and number of channels respectively. $M$ is the number of candidate answers. For multiple choice based selection tasks, $x_v, x_t$, and $Y$ are directly present in dataset. For N-Way Classification tasks, we set $x_t$ as a generic question (details in Appendix A) and formulate $Y$ by applying a fixed template to the labels of all N classes of the dataset. This formulation is used for the remainder of the paper unless a specific exception is noted.

In the case of open-ended video question answering, we follow standard settings of LLM based text generation for video tasks following Maaz et al. (2023).

Figure 2: **Likelihood Selection Workflow:** We illustrate how the likelihood selection strategy adapted for video QnA tasks can be efficiently parallelized (i.e. calculated with a simple cross-entropy loss in one forward pass, followed by an argmin operation), in contrast to the setting of iteratively generating multiple tokens.

## 3.2 LIKELIHOOD SELECTION

The common technique for LLM based approaches tackling question answering (QnA) tasks is likelihood based choice selection (also referred as Cloze Prompting, see Robinson et al. (2023)). Adopting such likelihood based selection for different tasks (or to VLMs) is however not straight-forward (Robinson et al., 2023), leading to most existing long video QnA approaches resorting to LLM based answer generation. In fact, most existing long-video QnA approaches using LLMs / VLMs for choice selection (Papalampidi et al., 2023; Wang et al., 2022; Balavzeví'c et al., 2024) resort to full answer generation followed by embedding or template based matching to ground-truth choices, incurring significant inference costs for evaluation.

In light of this, we explore prompting and likelihood calculation techniques optimal for applying *Likelihood Selection* on long video QnA tasks with either LLMs or even VLMs. Adapting this technique unlocks autoregressive LLMs / VLMs ability to solve multiple selection problems with only *one* forward pass as illustrated in Figure 2. This is in contrast to next token sampling requiring iterative generations dependent on previous outputs for each answer token. This process uses a likelihood measure based on the LLM latent space allowing better semantic awareness compared to exact or template matching. In addition to the candidate answer batching, we follow prior work to include all candidates in the prompt as well. We direct the reader to Table A.4 for complete details on semantic awareness, candidates in prompts, and video QnA specific implementation.

In addition to the considerable inference speed-up from likelihood selection, we also obtain the additional advantages of avoiding LLM hallucinations and deviations from expected output formats over iterative generation strategies applied to similar visual tasks (see Hanu et al. (2023)). We empirically validate the improved performance from such behavior in our ablations (see Table 6).

## 3.3 MODALITY CONSTRAINED VARIANTS

We next introduce the two modality-constrained variants of our framework tagged *Just-LLM* and *Single-Frame-VLM* (illustrated in Figure 1). The former utilizes only the task question injected as language ($x_t$) with no other task-specific information. Note how this naive variant does not access any information extracted from the video for each task instance. The latter utilizes an additional center visual frame ($x_v^c$), extracted from the center of the video ($x_v$) timeline. This variant accesses no *video-specific* data (e.g. temporal or motion information). The center frame usage ensures no temporal information leakage in frame selection for this variant.

We hypothesize that Just-LLM with no access to task-specific knowledge is constrained to generate predictions utilizing its internal world knowledge (e.g. physics, culture, human common sense). We refer to this as *world modality*. For a given question regarding a natural video and a set of candidate answers, there is a possibility that one choice is more probable given how our world operates. In cases that this choice turns out to be correct, the internal world knowledge of the LLM allows it to easily select that choice resulting in above random performance. This variant of our framework highlights such cases in long video QnA tasks. A similar baseline is used in Min et al. (2024).

In the case of Single-Frame-VLM, it is provided with task information but is limited to a single frame, which could possibly provide important scene context. Therein, we refer to this variant as operating with world and *contextual* information modalities. For example, consider a video with a man walking a dog. The scene context of the dog and man combined with the LLM world knowledge and reasoning skills may be sufficient to correctly answer the question with no temporal or motion

Table 1: **Modality Constrained Variants:** We report accuracy (%) and inference time per sample (s) on the public subset of EgoSchema (ES-S) and test set of NextQA (NextQA-T) datasets. Note that recent state-of-the-art from Zhang et al. (2023a) (SOTA) and our variants are implemented with common LLMs / VLMs and evaluated under identical settings.

| Method | Param | Video Frames | ES-S | | NextQA-T | |
|---|---|---|---|---|---|---|
| | | | Acc | Time | Acc | Time |
| Random | - | - | 20.0 | - | 20.0 | - |
| Just-LLM | 7B | 0 | 45.8 | 0.41 | 40.1 | 0.55 |
| SF-VLM | 13B | 1 | 55.8 | 1.89 | 51.2 | 2.03 |
| SOTA | 20B | 180 | 50.8 | 381 | 54.3 | 207 |

information. Performance of this variant highlights the prevalence of similar cases in long video QnA tasks when using LLM based approaches.

We evaluate these two modality-constrained variants and summarize our findings in Table 1. We uncover surprisingly strong performance of both variants on two long-video understanding benchmarks. In the case of Just-LLM variant, we achieve performance significantly higher than random selection (+25.8% on ES-S / +20.1% on NextQA-T) using zero visual information. This indicates the large portion of questions in existing video-QnA benchmarks that can be answered correctly purely using world knowledge. We also highlight our Single-Frame-VLM performing on par with state-of-the-art LLM based approach from Zhang et al. (2023a). In particular, for ES-S we outperform Zhang et al. (2023a) which uses information extracted from 180 frames per video incurring an inference cost over 100 times higher than ours. In light of these findings, we argue that long video understanding approaches in particular must focus on learning information beyond what a single frame baseline can achieve, possibly in an interpretable manner.

Therein, we introduce *Multimodal Video Understanding* (MVU), a simple framework that aggregates multimodal video-relevant information in an interpretable manner using natural language and achieves significant improvements over baselines across multiple datasets.

# 4 MULTIMODAL VIDEO UNDERSTANDING FRAMEWORK

In this section, we introduce in detail our Multimodal Video Understanding (MVU) framework that integrates several information modalities extracted from video using *natural language* as a medium for information fusion. Our approach adapts off-the-shelf vision tools to construct a powerful long video understanding agent that requires no additional training on videos. We first utilize vision tools to extract information relevant to three object-centric modalities from uniformly sampled video frames. Next, we leverage suitable prompt templates to aggregate these as natural language. This video level information is injected into our Single-Frame-VLM variant providing it with video specific awareness. We illustrate an overview of our framework in Figure 3.

## 4.1 VISION TOOLS FOR VIDEO ANALYSIS

Image trained VLMs contain information valuable for video tasks and have been widely used in prior work (Zhang et al., 2023a). In our proposed framework, we take a step further, exploring more off-the-shelf vision tools trained only on images, in particular object detection and object tracking approaches, in addition to a VLM re-purposed as an image captioner.

We use an image captioner to identify all unique objects present within a video. For this purpose, we prompt a generative vision language model to list all objects within a given video frame (image) in an open-ended manner. We note how a VLM trained only on images is sufficient for this. In our case, we use a VLM identical to the one in Zhang et al. (2023a) but applied on significantly less video frames, making our comparisons fair in terms of model size.

For the case of object detection, we use an open-vocabulary object detector from Minderer et al. (2022) that is trained only on images, and apply it with object category names from captioner to obtain their location information, i.e. image-space coordinates for each unique object. Given the lightweight nature of this detector in comparison to the image captioner, we note how it can be applied more densely (i.e. on more frames) than the captioner without increasing compute demand significantly. Furthermore, the detector acts as a secondary check, grounding the object category names to individual frames, and therein countering any object hallucinations by the captioner.

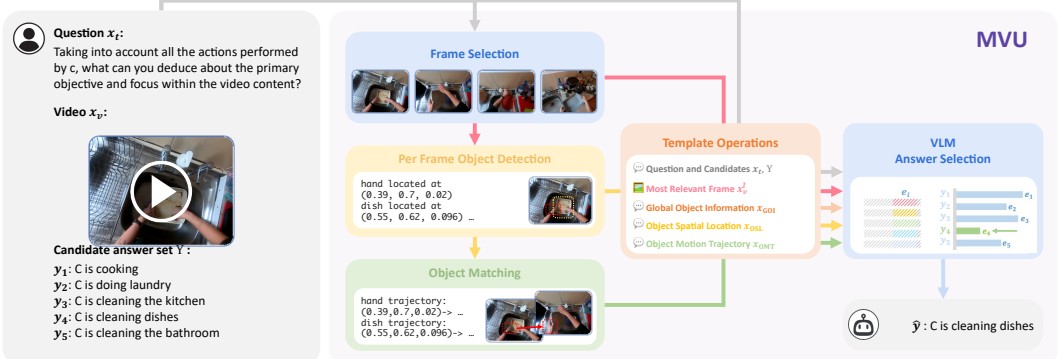

Figure 3: Overview of proposed framework for Multimodal Video Understanding, MVU.

Our final tool is an object tracker from Wang et al. (2018) used to convert our per-frame object detections into motion trajectories spread across the entire video. We feed the tracking algorithm with the locations of each object alongside per-object features extracted from our detector in order to construct motion trajectories for each unique object.

## 4.2 OBJECT-CENTRIC INFORMATION MODALITIES

Given off-the-shelf tools suitable for extracting information from videos, we next focus on the exact forms of information, i.e. three object-centric information modalities. We consider all object categories across the video, spatial locations of individual object instances, and their movement across time. We define these as follows:

1. **Global Object Information** ($x_{\text{GOI}}$): In this stage, we introduce global information that spans beyond a single video frame. For a given video, we first uniformly sample 8 frames. For each of the 8 selected frames, we utilize our image captioner to generate object lists and obtain a set of distinct object categories contained within each frame across the video.

2. **Object Spatial Location** ($x_{\text{OSL}}$): Given objects present per video, we utilize our open-vocabulary object detector to localize each object category (from previous stage) on to frame coordinates. Categories not localized by the detector are dropped. Additionally, we utilize similarity of feature vectors for same class objects to track object instances across frames using our tracker. Following prior work (Ranasinghe et al., 2024), we calculate average center coordinates and scale value for each object instance across all frames. This results in a set of distinct objects $O$ across the video, $O = \{(o_1, q_1), (o_2, q_2), ...\}$. Here, $o_k$ describes the object category in natural language while $q_k$ contains the x, y coordinates of object center and the scale term (area of minimal object bounding box as a ratio to image size, i.e. box area ÷ image size).

3. **Object Motion Trajectory** ($x_{\text{OMT}}$): Next, we leverage the calculated cross-frame object tracks and compute motion trajectories for each object. This modifies our set of distinct objects, pairing each object $o_k$ with its trajectory ($o_k^1 \to o_k^2 \to ...$) across the video frames. We construct an updated set $Z = \{(o_1, q_1^1 \to q_1^2 \to ...), (o_2, q_2^1 \to q_2^2 \to ...), ...\}$. Intuitively, this information should explicitly capture object motion information.

We provide further details including examples of each information modality for selected samples (video question pairs) in Appendix A.

This pipeline for extracting per-frame information using an image-trained VLM closely resembles prior work such as (Zhang et al., 2023a). While motivated by such work, we explore the direction of how more fine-grained information could be extracted from videos to solve these tasks more efficiently. Given the role of object interactions in defining the various actions and events in videos, we hypothesize that extracting object-centric information (as opposed to generic frame-level descriptions) followed by modeling of their temporal dependencies would provide more concise representations better suited to efficiently solve these tasks.

Table 2: **Ego-Schema Dataset Evaluation**: We report top-1 accuracy (%) for video question answering on Ego-Schema (Mangalam et al., 2023) test set (5031 videos). Our proposed MVU achieves state-of-the-art performance on this benchmark under *zero-shot operation with no video level training*. We also draw attention to our modality-constrained SF-VLM baseline that achieves surprisingly competitive performance.

| Method | Zero Shot | Video Training | Closed Model | Params | Full |
|---|---|---|---|---|---|
| Random Selection | - | - | - | - | 20.0 |
| VIOLET (Fu et al., 2022) | ✓ | ✓ | ✗ | 198M | 19.9 |
| FrozenBiLM (Yang et al., 2022) | ✓ | ✓ | ✗ | 1.2B | 26.9 |
| SeViLA (Yu et al., 2024) | ✓ | ✓ | ✗ | 4B | 22.7 |
| mPLUG-Owl (Ye et al., 2023b) | ✓ | ✓ | ✗ | 7.2B | 31.1 |
| InternVideo (Wang et al., 2022) | ✓ | ✓ | ✗ | 478M | 32.1 |
| ImageViT (Papalampidi et al., 2023) | ✗ | ✓ | ✗ | 1B | 30.9 |
| SeViLA+ShortViViT (Papalampidi et al., 2023) | ✗ | ✓ | ✗ | 5B | 31.3 |
| LongViViT (Papalampidi et al., 2023) | ✗ | ✓ | ✗ | 1B | 33.3 |
| MC-ViT-L (Balavzevi'c et al., 2024) | ✗ | ✓ | ✗ | 424M | 44.4 |
| InternVideo2 (Wang et al., 2024b) | ✓ | ✓ | ✗ | 7B | 55.8 |
| Tarsier (Wang et al., 2024a) | ✓ | ✓ | ✗ | 7B | 49.9 |
| Tarsier (Wang et al., 2024a) | ✓ | ✓ | ✗ | 34B | 61.7 |
| Vamos (Wang et al., 2023a) | ✓ | ✗ | ✗ | 13B | 36.7 |
| LLoVi (Zhang et al., 2023a) | ✓ | ✗ | ✗ | 13B | 33.5 |
| LangRepo (Kahatapitiya et al., 2024) | ✓ | ✗ | ✗ | 12B | 41.2 |
| Vamos (Wang et al., 2023a) | ✓ | ✗ | ✓ | 1.8T | 48.3 |
| LLoVi (Zhang et al., 2023a) | ✓ | ✗ | ✓ | 1.8T | 50.3 |
| LifelongMemory (Wang et al., 2023b) | ✓ | ✗ | ✓ | 1.8T | 62.4 |
| MoreVQA (Min et al., 2024) | ✓ | ✗ | ✓ | - | 51.7 |
| VideoAgent (Wang et al., 2025) | ✓ | ✗ | ✓ | 1.8T | 54.1 |
| VideoTree (Wang et al., 2024c) | ✓ | ✗ | ✓ | 1.8T | 61.1 |
| LVNet (Park et al., 2024) | ✓ | ✗ | ✓ | 1.8T | 61.1 |
| SF-VLM (ours) | ✓ | ✗ | ✗ | 13B | 36.4 |
| SF-VLM + MVU (ours) | ✓ | ✗ | ✗ | 13B | 37.6 |
| LVNet + MVU (ours) | ✓ | ✗ | ✓ | 1.8T | 61.3 |

## 4.3 LANGUAGE BASED FUSION

Inspired by Zeng et al. (2022), we construct our overall framework by injecting these three forms of object-centric information into our setup using natural language. We represent each modality in a fixed template-based fusion. Global object information is represented as a list of category labels, e.g., $x_{\text{GOI}} = \{person, oven, dishwasher, ..., sink\}$. Object spatial location modifies this list to include center coordinates $(x, y)$ and scale $(s)$ where scale is the area percentage occupied by the best-fitting object bounding box. For e.g., $x_{\text{OSL}} = \{person\ located\ at\ (0.2, 0.3, 0.07), ... , oven\ located\ at\ (0.8, 0.6, 0.04)\}$. Finally, object motion trajectories update the list to contain frame-level trajectories, e.g., $x_{\text{OMT}} = \{person\ moving\ as\ [0.2, 0.3, 0.07] \rightarrow [0.2, 0.4, 0.06] \rightarrow [0.2, 0.6, 0.08], oven\ moving\ as\ ...\}$. Similar to the examples, information from each object-centric modality is represented in textual form to allow their direct fusion and integration into our framework (as additional language inputs). Therein, we describe the resulting setup, our overall framework MVU as follows,

$$\hat{y} = \mathcal{F}_{\text{MVU}}(x_t, x_v^c, x_{\text{GOI}}, x_{\text{OSL}}, x_{\text{OMT}}) \tag{1}$$

where $x_v^c$ is the center frame extracted from the video $x_v$ (more details in Appendix A). In comparison to prior work such as Zhang et al. (2023a), we note that our fused information is more concise allowing better utilization of the fixed context length in an LLM (see Appendix K for more details).

## 5 EXPERIMENTS

In this section, we first discuss our experimental setup and datasets. Next, we evaluate MVU on multiple video question-answering and robotics task benchmarks followed by ablative studies.

**Experimental Setup:** Our proposed MVU framework and its variants use off-the-shelf models trained on images, thus requiring no re-training of these models. For our evaluations, we directly use these models, utilizing two NVIDIA RTX A5000 24GB GPUs for inference. We evaluate on two video question answering datasets focused on long-form videos: EgoSchema (Mangalam et al., 2023) and NExT-QA (Xiao et al., 2021). We also evaluate using a series of robotics datasets from

Table 3: **Next-QA Dataset Evaluation**: We report top-1 accuracy (%) for the Next-QA dataset (Xiao et al., 2021). Our proposed MVU achieves state-of-the-art results under zero-shot settings with *no video-level training*. In table header, ZS stands for zero-shot and VT stands for video level training.

| Method | ZS | VT | Params | Cau. | Tem. | Des. | All |
|---|---|---|---|---|---|---|---|
| Random Selection | - | - | | 20.0 | 20.0 | 20.0 | 20.0 |
| CoVGT (Xiao et al., 2023) | ✗ | ✓ | 149M | 58.8 | 57.4 | 69.3 | 60.0 |
| SeViT (Kim et al., 2023) | ✗ | ✓ | 215M | - | - | - | 60.6 |
| HiTeA (Ye et al., 2023a) | ✗ | ✓ | 297M | 62.4 | 58.3 | 75.6 | 63.1 |
| InternVideo (Wang et al., 2022) | ✗ | ✓ | 478M | 62.5 | 58.5 | 75.8 | 63.2 |
| MC-ViT-L (Balavzevi'c et al., 2024) | ✗ | ✓ | 424M | - | - | - | 65.0 |
| BLIP-2 (Li et al., 2023a) | ✗ | ✓ | 4B | 70.1 | 65.2 | 80.1 | 70.1 |
| SeViLA (Yu et al., 2024) | ✗ | ✓ | 4B | 74.2 | 69.4 | 81.3 | 73.8 |
| LLama-VQA-7B (Ko et al., 2023) | ✗ | ✓ | 7B | 72.7 | 69.2 | 75.8 | 72.0 |
| Vamos (Wang et al., 2023a) | ✗ | ✓ | 7B | 72.6 | 69.6 | 78.0 | 72.5 |
| Just-Ask (Yang et al., 2021) | ✓ | ✓ | 66M | 31.8 | 30.4 | 36.0 | 38.4 |
| VFC (Momeni et al., 2023) | ✓ | ✓ | 164M | 45.4 | 51.6 | 64.1 | 51.5 |
| InternVideo (Wang et al., 2022) | ✓ | ✓ | 478M | 43.4 | 48.0 | 65.1 | 49.1 |
| SeViLA (Yu et al., 2024) | ✓ | ✓ | 4B | 61.3 | 61.5 | 75.6 | 63.6 |
| CaKE-LM (Su et al., 2023) | ✓ | ✗ | 2.7B | 35.7 | 35.3 | 36.8 | 34.9 |
| LLoVi (Zhang et al., 2023a) | ✓ | ✗ | 13B | 55.6 | 47.9 | 63.2 | 54.3 |
| ViperGPT (Surís et al., 2023) | ✓ | ✗ | 175B | - | - | - | 60.0 |
| LLoVi (Zhang et al., 2023a) (GPT-4) | ✓ | ✗ | 1.8T | 69.5 | 61.0 | 75.6 | 67.7 |
| MoreVQA (Min et al., 2024) | ✓ | ✗ | 1.7T | 70.2 | 64.6 | - | 69.2 |
| VideoAgent (Wang et al., 2025) | ✓ | ✗ | 1.7T | 72.7 | 64.5 | 81.1 | 71.3 |
| VideoTree (Wang et al., 2024c) | ✓ | ✗ | 1.7T | 75.2 | 67.0 | 81.3 | 73.5 |
| LVNet (Park et al., 2024) | ✓ | ✗ | 1.8T | 75.0 | 65.5 | 81.5 | 72.9 |
| SF-VLM + MVU (ours) | ✓ | ✗ | 13B | 55.7 | 48.2 | 64.2 | 55.4 |
| LVNet + MVU (ours) | ✓ | ✗ | 1.8T | 75.2 | 66.8 | 81.3 | 73.3 |

the Open X-Embodiment robotics dataset (Open-X-Embodiment-Collaboration et al., 2023) to test our model generality (more details in Section 5.2). We discuss further details of pretrained models and datasets in Appendix B. Also, note that none of the pretrained components of our framework undergo any form of video-level training.

## 5.1 LONG VIDEO QUESTION ANSWERING

Long video question answering benchmarks aim to measure causal and temporal reasoning abilities of models over long temporal windows (Xiao et al., 2021; Mangalam et al., 2023). In this section, we evaluate our framework on two benchmark datasets and present our results in Table 2 and Table 3.

On EgoSchema dataset, results reported in Table 2 demonstrate the state-of-the-art performance of our framework. We integrate MVU over SF-VLM and LVNet (Park et al., 2024) baselines for fair comparison to work operating under different settings. We reiterate how our approach is both zero-shot and requires no video-level training (and our selected baselines are similar). In comparison to prior work utilizing open models, our SF-VLM+MVU achieves clear performance improvements, even out-performing works using video-caption supervision for training (Papalampidi et al., 2023; Balavzevi'c et al., 2024). Compared to methods utilizing proprietary closed language models extending to trillion parameter scale (Zhang et al., 2023a; Wang et al., 2023a; Min et al., 2024; Wang et al., 2025), our LVNet+MVU variant using similar scale achieves improved performance. We also implement several such large-scale approaches under scaled-down common settings as our smaller variant (details in Appendix C), where we again achieve clear performance gains.

Next, we evaluate our framework on the NextQA benchmark and report these results in Table 3. We similarly integrate MVU with two baselines. Our MVU achieves state-of-the-art results under zero-shot settings. While Yu et al. (2024) outperforms our approach, we note how they require video-caption localization pretraining and appears to overfit to this dataset considering their relatively lower performance on other datasets (see Table 2).

We also evaluate MVU on the LongVideoBench dataset which contains even longer videos and present these results in Appendix H. While these three datasets focus on MCQ style QnA, we also explore the generality of our MVU framework on open-ended style QnA tasks in Appendix G.

Table 4: **OpenX Detailed Results:** We report accuracy (%) for the VideoQA formulation of Open X-Embodiment benchmark. MVU achieves clear improvements over random selection and LLoVi baseline (Zhang et al., 2023a). In table header, Obs. (observation), size, CC (class count) stand for camera used, number of videos, and number of unique language instructions per dataset, respectively. In observation column, T stands for third-person view (stationary camera that does not move with robot), while F denotes first-person view where camera is mounted on moving robot. Note that total is average weighted by dataset size.

| Dataset | Obs. | Size | CC | Random | Baseline | MVU |
|---|---|---|---|---|---|---|
| ASU TableTop Manipulation | T | 110 | 83 | 13.6 | 19.1 | **20.9** |
| Berkeley MVP Data | F | 480 | 6 | 20 | 26.0 | **33.1** |
| Berkeley RPT Data | F | 908 | 4 | 24.6 | 23.1 | **26.2** |
| CMU Play Fusion | T | 576 | 44 | 20.3 | 34.0 | **35.6** |
| CMU Stretch | T | 135 | 5 | 23 | 18.5 | **24.4** |
| Furniture Bench | T | 5100 | 9 | 20.2 | 24.8 | **26.4** |
| Furniture Bench | F | 5100 | 9 | 20.2 | 22.6 | **24.9** |
| CMU Franka Pick-Insert Data | T | 631 | 7 | 18.7 | 19.3 | **21.2** |
| CMU Franka Pick-Insert Data | F | 631 | 7 | 23.1 | **57.8** | 49.3 |
| Imperial F Cam | T | 170 | 17 | 20 | 22.9 | **24.1** |
| Imperial F Cam | F | 170 | 17 | 23.5 | 20.6 | **24.7** |
| USC Jaco Play | T | 1085 | 89 | 21.8 | 26.4 | **30.6** |
| USC Jaco Play | F | 1085 | 89 | 19.4 | 28.6 | **32.4** |
| NYU ROT | T | 14 | 12 | 21.4 | **57.1** | **57.1** |
| Roboturk | T | 1959 | 3 | 34.7 | 43.0 | **44.2** |
| Stanford HYDRA | T | 570 | 3 | 35.1 | 54.7 | **68.2** |
| Stanford HYDRA | F | 570 | 3 | 31.2 | 45.3 | **48.9** |
| Freiburg Franka Play | F | 3603 | 406 | 20.4 | **32.2** | 31.6 |
| Freiburg Franka Play | T | 3603 | 406 | 19.7 | 21.8 | **24.0** |
| LSMO Dataset | T | 50 | 2 | 34.0 | 68.0 | **72.0** |
| UCSD Kitchen | T | 150 | 8 | 19.3 | 32.0 | **32.7** |
| Austin VIOLA | T | 150 | 3 | 26.7 | 32.7 | **33.3** |
| Austin VIOLA | F | 150 | 3 | 30.0 | 33.3 | **34.0** |
| Total | - | 27000 | - | 22.1 | 28.5 | **30.4** |

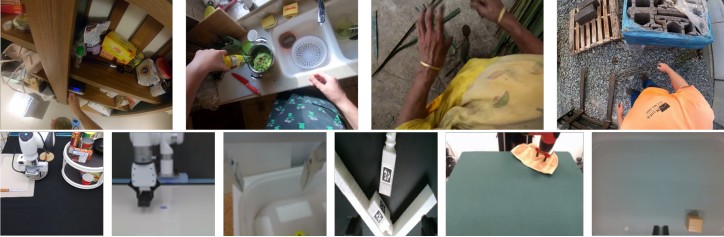

Figure 4: **Data Visualization:** Example video frames from EgoSchema (top) vs OpenX (bottom) datasets. Robotics domain videos (bottom) appear out of distribution given their controlled environment and robot movements.

## 5.2 ROBOTICS DOMAIN ACTION RECOGNITION

We investigate generalization capabilities of our proposed MVU by evaluating across datasets from robotics domain Open X-Embodiment (Open-X-Embodiment-Collaboration et al., 2023), following a QnA style formulation of the dataset (details in Appendix D). We highlight visual differences of this data in Figure 4. We present evaluations in Table 4, which indicate clear performance improvements for MVU over the baseline from Zhang et al. (2023a). The purpose of this experiment is to evaluate the generality of our approach to video domains different from everyday natural videos. We take these promising results to indicate the strong generality of our framework. Furthermore, we note how our modality constrained variants do not perform significantly better than random on these robotics domain tasks (details in Appendix E). We attribute this to the significant domain shift in terms of the world of operation in this domain (i.e. robotics tasks tend to involve controlled environments very different to what humans face on an everyday basis).

## 5.3 ABLATIONS

In this section, we systematically dissect our overall MVU framework to establish the usefulness of each of its individual component (see Appendix I for more ablations). We first ablate our three different information modalities and report these results in Table 5. Our evaluations indicate clear performance improvements from each of our object-centric information modalities.

Table 5: **MVU Ablation:** We report accuracy (%) on public subset of EgoSchema (ES-S). In table header, VI stand for visual inputs and GOI, OSL, OMT refer to our object-centric information modalities (see Section 4.2). Each information modality results in clear performance improvements with our full MVU achieving best performance.

| Method | VI | GOI | OSL | OMT | ES-S |
|---|---|---|---|---|---|
| Just-LLM (ours) | ✗ | ✗ | ✗ | ✗ | 45.8 |
| SF-VLM (ours) | ✓ | ✗ | ✗ | ✗ | 55.8 |
| MVU (ours) | ✓ | ✓ | ✗ | ✗ | 56.4 |
| MVU (ours) | ✓ | ✓ | ✓ | ✗ | 58.6 |
| MVU (ours-full) | ✓ | ✓ | ✓ | ✓ | **60.3** |

Table 6: **Likelihood Selection (LS) Ablation:** Results indicate clear improvements in both accuracy (%) and inference time (s) with our adaptation of likelihood selection for video tasks.

| Method | LS | ES-S | NQA-T | Time |
|---|---|---|---|---|
| Generation | ✗ | 56.4 | 55.3 | 12.7 |
| LS-Naive | ✗ | 58.2 | 35.8 | 2.42 |
| LS-MVU (ours) | ✓ | **60.3** | **55.4** | **2.42** |

Table 7: **Baseline Ablation:** We replace information input to final stage VLM with frame descriptions following Zhang et al. (2023a). Accuracy (%) on public subset of EgoSchema (ES-S). Time in seconds (s).

| Method | Frames | ES-S | Time |
|---|---|---|---|
| Baseline | 180 | 55.4 | 207 |
| Baseline | 8 | 55.8 | 2.38 |
| Baseline | 16 | 56.2 | 4.72 |
| MVU (ours) | 16 | **60.3** | 2.42 |

We next perform ablations on our adaptation of likelihood selection strategy for video QnA tasks using Ego-Schema subset (ES-S) and Next-QA test-set (NQA-T) . These results reported in Table 6 indicate clear performance boosts due to our adaptation of likelihood selection (LS). When removing LS, standard generation (i.e. generate an answer and match against ground-truth selection following Zhang et al. (2023a)) is utilized with our MVU framework. We also report naive adaptation of LS following Robinson et al. (2023) where the choice options are directly used, highlighting the importance of our prompting techniques. We also note the accuracy gains obtained through LS, and attribute these to reduced LLM output hallucination and deviations from expected output formats, that are commonly observed with iterative generation (Hanu et al., 2023).

We next ablate our overall framework against the existing work, Zhang et al. (2023a), by replacing our MVU object-centric information pipeline with the frame description approach in Zhang et al. (2023a). We construct a setup identical to our framework except for the inputs to our final stage VLM replaced with frame level descriptions. These results reported in Table 7 indicate the clear significance and improvement of our proposed object-centric information pipeline over simple frame descriptions. The 8 frame variant is the same speed comparison as MVU uses captioner only on 8 frames. Our MVU outperforms both that and the slower 16 frame baseline. We also note the performance drop in the baseline when increasing the number of frames from 16 to 180. While consistent with observations in prior works for long-video tasks (Mangalam et al., 2023), we attribute this drop to decreased signal-to-noise ratio with the introduction of additional frame descriptions. This further highlights the importance of selecting useful information from video frames, and we reiterate how the object-centric information in our MVU framework serves this purpose.

## 6 CONCLUSION

In this work, we present a multimodal video understanding framework, termed MVU, that achieves state-of-the-art performance on complex video understanding tasks. In particular, evaluations on long-video question answering and robotics domain question answering demonstrate the strong performance of our MVU framework as well as its generality. We also adapt likelihood selection for efficient LLM-based answer choice selection, separate video-specific information into three object-centric modalities, demonstrate automated extraction of such information using off-the-shelf vision tools, and propose language-based fusion of this multimodal information.

We also presented two modality-constrained baselines that uncover surprising insights relevant to LLM based video QnA which serves as a basis for our subsequent MVU framework. Furthermore, these results highlight the need for careful evaluation of LLM-based video QnA approaches. Revisiting our original motivation on *"what we can learn from videos, beyond scene context understood from a single natural image"*, in this work our two modality-constrained variants uncover surprising insights relevant to this question. We first achieve strong results on long-video understanding benchmarks using no video-specific data, and build over that baseline to showcase the additional performance gains achievable through injecting video-specific information.

## REPRODUCIBILITY STATEMENT

Our method utilizes multiple pretrained models, all of which are open-source with model weights freely available. We use the versions of these models hosted on HuggingFace https://huggingface.co for all our experiments. For state-of-the-art baseline LVNet, we utilize code from https://github.com/jongwoopark7978/LVNet. We discuss all steps in our proposed algorithms in detail while also releasing relevant code. All evaluations we perform are on public datasets accessible by all (some behind evaluation servers to prevent test set contamination). Our code is also publicly available at https://github.com/kahnchana/mvu.

## CONTRIBUTIONS

KR led the project by building the preliminary ideas followed by performing most experiments and evaluations. XL contributed to ideas on off-the-shelf model usage, setup all robotics evaluations, debugged many issues, and discussed all aspects of the project. KK contributed to ideas on VLM prompting and templating of textual data, setup several baseline replications, reviewed code, and discussed all aspects of the project. MR organized the project, set the research direction, and discussed all aspects of the project idea, scope, and implementation.

## ACKNOWLEDGMENTS

This work was supported by Electronics and Telecommunications Research Institute (ETRI) grants funded by the Korean government [24ZR1100, A Study of Hyper-Connected Thinking Internet Technology by autonomous connecting, controlling and evolving ways]. This work was also supported by the Institute of Information & Communications Technology Planning & Evaluation (IITP) grant funded by the Korea government(MSIT) (No. RS-2024-00336738, Development of Complex Task Planning Technologies for Autonomous Agents).

We thank all members of the Robot Learning Lab at Stony Brook University for support, feedback and guidance. In particular, we would like to thank Jongwoo Park for technical feedback and support. We also thank Si Chen for logistical help and encouragement throughout the project.

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

# Appendix

## A PROMPTING AND TEMPLATE OPERATIONS

In Section 4.2 and Section 4.3, we utilize 3 distinct prompts and fusion templates for generating joint textual inputs to be processed by the LLM. The 3 distinct prompt categories correspond to our Global Object Information ($x_{\text{GOI}}$), Object Spatial Location ($x_{\text{OSL}}$), and Object Motion Trajectory ($x_{\text{OMT}}$) modalities. We first describe our exact templates as Python pseudo-code in Table A.1.

---

**Global Object Information ($x_{\text{GOI}}$)**
```
"Consider following objects in video to answer the question:" + \
", ".join(GOI_data) + ".  " + task_question
```

**Object Spatial Location ($x_{\text{OSL}}$)**
```
"Consider following objects with spatial location as
(x, y, area) coordinates in video to answer the question:" + \
", ".join(OSL_data) + ".  " + task_question
```

**Object Motion Trajectory ($x_{\text{OMT}}$)**
```
"Consider following objects moving along (x, y, area) trajectories
in video to answer the question:" + \
", ".join(OMT_data) + ".  " + task_question
```

Table A.1: **Prompt templates for three textual modalities.**

The above templates depend on each of the modalities represented in textual form (i.e. `GOI_data`, `OSL_data`, `OMT_data`). We describe their exact textual forms next using examples in Table A.2.

---

```
GOI_data = ["person", "oven", "dishwasher", "sink", "countertop",
"dish", "box", "scissors", "drain", "hand", "stove"]
```

```
OSL_data = ["stove located at (0.52, 0.64, 0.595)",
        "sink located at (0.56, 0.64, 0.211)",
        "countertop located at (0.63, 0.79, 0.308)",
        "box located at (0.46, 0.65, 0.142)",
        "dishwasher located at (0.5, 0.5, 0.991)",
        "dish located at (0.41, 0.75, 0.077)",
        "person located at (0.47, 0.76, 0.282)" ]
```

```
OMT_data = ["stove trajectory:  (0.5,0.5,0.991)->(0.51,0.69,0.397)
        ->(0.54,0.73,0.396)",
        dish trajectory:  (0.55,0.62,0.096)->(0.11,0.65,0.079)",
                            .
                            .
                            .
        "dish trajectory:  (0.41, 0.75, 0.077)",
        "person trajectory:  (0.54,0.81,0.34)->(0.49,0.72,0.339)
        ->(0.54,0.84,0.157)->(0.23,0.71,0.176)
        ->(0.51,0.79,0.232)->(0.52,0.78,0.266)
        ->(0.39,0.64,0.558)->(0.54,0.82,0.184)"]
```

Table A.2: **Prompt examples for three textual modalities.**

In this example (for a single video), the `GOI_data` list contains 11 distinct object categories discovered across all 8 selected frames for this video. In `OSL_data`, this category list is grounded to each frame using our object detector. We apply this on 16 uniformly sampled frames as opposed to only 8 used with the captioner. While this stage removes some categories (which we assume could be object hallucinations (Ranasinghe et al., 2024)), it also identifies categories at the instance

level. We draw attention to the two different instances of a dish in our `OSL_data` for this example. Also, note that the single spatial coordinate reflects the average location of that object across all 16 (or the number of frames it is detected in) following the setup in Ranasinghe et al. (2024). Our object tracks calculated across frames are utilized for this averaging (i.e. distinguish the two dishes). For our `OMT_data`, we again leverage our tracks where each object instance is matched across frames and construct explicit sequences of object locations across frames. While ignoring the actual frame indexes, we only consider the object trajectory using frames where they are detected. Note that an object trajectory could be limited to a single location or a variable number of multiple locations. Also, for these trajectories, we introduce an additional scale factor for each object location. This scale factor is the ratio of the object bounding box area to image area, i.e. $(\mathtt{obj\_width} * \mathtt{obj\_height}) \div \mathtt{im\_size}$. This is introduced with an aim of possibly providing some level of depth information. In terms of generating object tracks, we utilize intermediate features from our object detector and perform feature matching based tracking.

## B  DETAILS ON PRETRAINED MODELS AND DATASETS

We describe in detail the pretrained models used to construct our framework as well as the multiple datasets used in evaluating our framework.

**Models:** Our framework utilizes three distinct off-the-shelf models for its various operations, namely *a*) an LLM / VLM for likelihood selection, *b*) a generative VLM for extracting object list from a frame, and *c*) an open-vocabulary detector for object localization. We use `LLaVA-v1.5-13B` (Liu et al., 2023a) for likelihood selection and frame object list generation. For object localization, we use `OWL-ViT-B/32` (Minderer et al., 2022). Unless explicitly specified, we use the above setup in all our experiments. Variants of our framework uses LLMs `Llama-2-7b-Chat`, `Gemma-7b-IT`, and `Mistral-7B-Instruct` (default) for likelihood selection. Apart from these off-the-shelf models, our framework involves zero additional training. We also reiterate that no components of our framework undergo any form of video-level training.

**Datasets:** We use multiple video datasets for evaluation under question-answering or n-way classification settings. For video question answering, we select two datasets focused on long-form videos: EgoSchema (Mangalam et al., 2023), NExT-QA (Xiao et al., 2021). EgoSchema is a long-form egocentric video question-answering benchmark, consisting of a 500-video public subset (EgoSchema-S) and a full 5000+ video evaluation set (EgoSchema-F) accessed only through evaluation servers. This dataset spans over 250 hours and is specially constructed to ensure that *questions require awareness of a longer temporal window for correctly answering* (Mangalam et al., 2023). Example images of EgoSchema are shown in Figure 4. NExT-QA similarly contains long-form videos with a focus on requiring causal & temporal action reasoning as well as common scene comprehension for correctly answering. It contains a validation set (NExT-QA-V) of 4996 video-questions pairs and a test set (NExT-QA-T) of 8564 video-question pairs. We also use a series of robotics datasets from the Open X-Embodiment robotics dataset (Open-X-Embodiment-Collaboration et al., 2023) for video question answering in a different domain (more detail in Section 5.2). In only one of our ablations aimed at analyzing the motion understanding aspect of our framework, we utilize a fine-grained action recognition dataset, Something-Something-v2 (Goyal et al., 2017), that contains 174 action categories focusing on object motions by replacing object category nouns with '*something*' in textual descriptions of each action category.

## C  DETAILS ON BASELINES

In Section 5.1, we evaluate performance on long-video understanding tasks using work in Zhang et al. (2023a) and Wang et al. (2023a) as two baselines for comparison. However, both these methods utilize closed-source, proprietary LLMs (i.e. GPT-4) with parameter counts on the order of trillions (over 100X our model size) deeming their direct comparisons unfair. In the case of Zhang et al. (2023a), we replicate their method (using their open-source repository and pre-trained models following Kahatapitiya et al. (2024)) utilizing an open-source LLM of comparable parameter count as our framework. For Wang et al. (2023a), the authors directly report results for a variant with a similar parameter count as ours. We utilize these evaluations as our point of comparison.

We also replicate prior work, LVNet (Park et al., 2024), that exhibits state-of-the-art results. For this, we use their official code (https://github.com/jongwoopark7978/LVNet) and integrate our MVU framework over this baseline.

We highlight that re-implementations of these baselines utilize common LLMs / VLMs as our MVU framework followed by identical evaluation protocols to ensure fair comparison.

## D ROBOTICS DOMAIN DATASET DETAILS

The Open X-Embodiment dataset is an extensive collection of visuomotor robotics datasets, encompassing a wide range of tasks and environments. It is designed to facilitate research in visuomotor control, providing rich sensory inputs and motor outputs for training and testing robotic systems. However, the videos are usually taken in a controlled environment and they do not always contain meaningful objects, which makes the samples in the dataset usually out of general video distribution (See Figure 4).

For our analysis, we specifically select datasets within this collection that contain expert episodes accompanied by corresponding language instructions and adapt them into video classification datasets. We treat each trajectory within the dataset as a video clip, with its associated language instruction serving as the video caption (classification label). For each episode, the model is tasked with identifying the correct language instruction from a set of five options, considering a video clip uniformly downsampled to 8 frames. The incorrect options are randomly chosen from the dataset to ensure a diverse and challenging selection. In instances where the datasets have multiple cameras for each episode, we treat the videos captured by each camera as distinct datasets.

## E DISCUSSION ON MODALITY CONSTRAINED EVALUATION

We evaluate the two modality-constrained variants of our approach, Just-LLM and Single-Frame-VLM (details in Section 3.3) and summarize these findings in Table 1. We uncover surprisingly strong performance of both variants on two long-video understanding benchmarks. Note how these approaches use no video-specific information to generate predictions.

We highlight how our best Just-LLM variant achieves performance significantly higher than random selection (+25.8% on EgoSchema-S / +20.1% on NextQA-T) using zero visual information. This indicates the large portion of questions in existing video-QnA benchmarks that can be answered correctly purely using world knowledge. We also highlight our single frame variant performing on par with some existing state-of-the-art (gray). In particular, for EgoSchema-S we outperform Zhang et al. (2023a) which uses information extracted from 180 frames per video incurring an inference cost over 100 times higher than ours. In light of these findings, we argue that long video understanding approaches in particular must focus on learning information beyond what a single frame baseline can achieve.

We also evaluate these same modality-constrained variants on robotics domains tasks and report these results in Table A.3. In contrast to the results on standard long-video QnA benchmarks, the robotics domains results are more aligned with intuition: the no-visual input Just-LLM performs on par with random and the Single-Frame-VLM marginally outperforms random selection.

We attribute this difference in performance to the nature of robotics domain tasks. They tend to involve controlled environments with often naive, meaningless tasks purely for robot testing purposes. These may not necessarily align with human commonsense or other constraints dependent on knowledge of our world. Therein, the clear ability of LLMs to solve general everyday video tasks (e.g. EgoSchema, NextQA performance in Table 1) using its world knowledge may not be applicable to robotics domain tasks. Utilizing different domain benchmarks, in particular robotics tasks, provides a much more representative evaluation of LLM based video QnA approaches.

## F LIKELIHOOD SELECTION

In this section, we present the prompts and templates used to adapt likelihood selection inference (Robinson et al., 2023) to our video QnA tasks. Our experimentation shows significantly higher

Table A.3: **Modality Constrained Variants on Robotics Domain:** We evaluate our modality constrained baselines on the robotics domain tasks and report accuracy (%). Note that a weighted sum over multiple tasks is reported here (similar to Table 4). Note the minimal increase over random for the variants in contrast to generic video benchmarks.

| Method | Visual | Frames | Accuracy |
|---------|--------|--------|----------|
| Random | - | - | 22.1 |
| Just-LLM | ✗ | - | 21.9 |
| SF-VLM | ✓ | 1 | 23.5 |
| MVU | ✓ | 16 | 30.4 |

sensitivity (to prompt style) of LLM performance on QnA tasks when using like likelihood selection in comparison to sequential text generation (consistent with findings in Robinson et al. (2023)). We evaluate a series of different prompt templates on the EgoSchema and Next-QA dataset to discover optimal combinations. The best prompt templates used in our final framework are presented in Table A.4 as Python pseudo-code. For Next-QA in particular, the average zero-shot accuracy could vary from 35% to 55% with slight variations of the prompt templates.

Our optimal prompt templates for the standard video QnA tasks also generalized to our robotics domain QnA tasks. Nevertheless, we highlight the possibility of needing some prompt template tuning when applying our framework to different domains. We also note that while our prompt selection process was guided by heuristics and experimentation, there may be other similar prompts performing equally well or surpassing our optimal selection.

### F.1 IMPLEMENTATION DETAILS

We revisit the generation process of autoregressive LLMs and their visual extensions (VLMs). They commonly use iterative prediction of next tokens conditioned on prior outputs to generate complete natural language outputs. Such a generation process is usually modeled as sampling from a conditional likelihood shown as Equation (2), where $\hat{y}^j$ stands for the $j^{\text{th}}$ token in a textual sequence $\hat{y}$ autoregressively generated by the model.

$$P(\hat{y}|x_t) = \prod_j P(\hat{y}^j|\hat{y}^{1,\dots,j-1}, x_t) \tag{2}$$

The dependency on prior output $\hat{y}^{1,\dots,j-1}$ makes this process both computationally costly and redundant in the case of choice-based answer selection tasks. Alternately, given the closed set of $Y$ in choice-based selections tasks, we formulate $P(y_i|x_t)$ for any $y_i \in Y$ with no dependency on any model generated output ($\hat{y}$) as,

$$P(y_i|x_t) = \prod_j P(y_i^j|y_i^{1,\dots,j-1}, x_t) \tag{3}$$

Assume a perfect LLM, intuitively when $y_i$ is a proper answer to the question $x_t$ (say $y_i = y_g$), the conditional likelihood $P(y_i|x_t)$ should be larger than any other $P(y_w|x_t)$ where $y_w$ is a wrong answer to question $x_t$. In fact, modern LLMs are trained with a similar objective (Radford & Narasimhan, 2018). Motivated by this standard LLM training objective, we estimate the relative numerical scales of conditional likelihood on different answers $P(y_i|x_t)$ using a cross-entropy error $e_i$, given their equivalence (negative log-likelihood and multiclass cross-entropy, see Section 4.3.4 in Bishop (2006)). We calculate $e_i$ with a single forward pass of LLM without detokenization and the selection can be made by simply picking up the answer with the lowest error, equivalent to the highest conditional likelihood among all the answers.

This sets the ground for *Likelihood Selection*, also referred to as Cloze Promting in Robinson et al. (2023), first illustrated with a toy example in Figure 2, where the task is vanilla question-answering with only textual context and the model takes one question $x_t$ as well as $M = 5$ candidate answers $y_{1,\dots,5}$. To find the best answer, we simply concatenate the question with each candidate independently ($s_i = \text{concat}(x_t, y_i)$) and pad them into a batch $\{s_{1,\dots,5}\}$. Then the LLM takes the batch of five samples with causal attention masks and performs one inference forward pass, resulting in five shifted logits $\{p_{1,\dots,5}\}$. Next, we shift the input sequence $s_i$ to align the logits $p_i$ and calculate the average cross-entropy error only on tokens of $y_i$ Finally, the answer with the smallest $e_i$ will be picked up as the best answer. The method can be formulated as in Equation (5) using equivalence

```
prompt_list = \
    [f"Response {idx}:{val}" for idx, val in
enumerate(prompt_list)]

system_prompt = \
    "Considering given frames of a long video, select the
most
    suitable response to the following question from the
five
    options provided."

response_template = \
    "The correct response best answering the question about
the given
    video is "

task_prompt = "Question:  {qs}" + ''.join(prompt_list)

qs = system_prompt + task_prompt + response_template
```

Table A.4: **Likelihood Selection Sample Prompt Templates.** Variables *qs* and *prompt_list* refer to per sample question and choice list respectively.

of negative log-likelihood to cross-entropy in Equation (4). Here $n_i$ stands for the token sequence length of $y_i$ and $p_i^j$ stands for logits of the $j^{\text{th}}$ token in $p_i = V(\text{concat}(x_t, y_i))$ with logits limited to only those of $y_i$.

$$e_i(y_i) = \text{CE}(p_i, y_i) = \frac{1}{n_i} \sum_j^{n_i} \left( \text{CE}(p_i^j, y_i^j) \right) \approx \sum_j^{n_i} -\log P(y_i|x_t) \tag{4}$$

$$\mathcal{F}_{\text{LS}}(Y, x_t) = \arg\max_{y_i \in Y} P(y_i|x_t) = \arg\min_{y_i \in Y} e_i(y_i) \tag{5}$$

In summary, Likelihood Selection performs one single forward pass to extract the network logit outputs, calculates error ($e_i$) on each choice, and selects the choice with the lowest error. Note that our method does not utilize any iterative autoregressive generation using the LLM. This results in considerable speed-ups for inference time. We also obtain the additional advantages of avoiding LLM hallucinations and deviations from expected output formats over iterative generation strategies applied to similar visual tasks (Hanu et al., 2023) leading to better accuracy (see Tab. 6.). In Section 3.3, we demonstrate both our speed gains and performance improvements.

Furthermore, Likelihood Selection is a generic method that can be easily extended to autoregressive VLMs, and in principle, there is no reason it could not also be used with extra modalities besides language. We validate this across all our experiments using the multimodal MVU framework.

## F.2 DISTINCTION FROM EXACT MATCH

As described in the previous section, likelihood selection uses a likelihood measure which is the likelihood (probability) of the model generating the given sentence (as opposed to being an exact match). This likelihood measure is also used as the training loss when training LLMs. Given how LLMs trained with this loss (i.e. all decoder based LLMs such as LLaMA, Gemini, GPT) are highly effective at handling semantic meaning, it follows that this loss can capture semantic meaning. This likelihood measure is calculated within the LLM latent space. This is equivalent to the probability (or likelihood) of that answer being generated by the LLM conditioned on the input question. We derive this in detail in Appendix F. Relating to the same example, this means that likelihood is an estimate of how likely the model would predict 'C is washing plates' as opposed to making that

Table A.5: **Ablating Answer Candidates in Prompt:** We illustrate the importance of appropriate prompting when combining with likelihood selection, specifically for long video QnA tasks. Top-1 accuracy (%) is reported on EgoSchema subset (ES-S) and NextQA test set (NQA-T).

| Dataset | ES-S | NQA-T |
|---|---|---|
| No answer candidates in prompt | 58.2 | 35.8 |
| With answer candidates in prompt | 60.3 | 55.4 |

Table A.6: **Open-Ended Video QnA Evaluation**: We present results on the ActivityNet dataset (Yu et al., 2019) that demonstrate strong performance of our proposed MVU framework. Accuracy (%) is reported. VT stands for video level training. We highlight how our MVU framework utilizes no video level training for any of its components and surpassed multiple approaches that rely on video-language training.

| Method | Zero-Shot | VT | ActivityNet-QA |
|---|---|---|---|
| JustAsk (Yang et al., 2021) | ✗ | ✓ | 38.9 |
| FrozenBiLM (Yang et al., 2022) | ✗ | ✓ | 43.2 |
| VideoCoCa (Yan et al., 2022) | ✗ | ✓ | 56.1 |
| FrozenBiLM (Yang et al., 2022) | ✓ | ✓ | 24.7 |
| Video Chat (Li et al., 2023b) | ✓ | ✓ | 26.5 |
| LLaMA Adapter (Zhang et al., 2023c) | ✓ | ✓ | 34.2 |
| Video LLaMA (Zhang et al., 2023b) | ✓ | ✓ | 12.4 |
| Video-ChatGPT (Maaz et al., 2023) | ✓ | ✓ | 35.2 |
| LocVLM (Ranasinghe et al., 2024) | ✓ | ✓ | 37.4 |
| Video-LLaVA (Lin et al., 2023a) | ✓ | ✓ | 37.4 |
| VISTA-LLaMA (Ma et al., 2023) | ✓ | ✓ | 37.4 |
| VideoChat-2 (Li et al., 2023c) | ✓ | ✓ | 37.4 |
| LLaMa-VID (Li et al., 2023e) | ✓ | ✓ | 37.4 |
| LLoVi (Zhang et al., 2023a) | ✓ | ✗ | 41.8 |
| MVU (ours) | ✓ | ✗ | **42.2** |

exact match. This means predictions closer to the target such as 'C is cleaning dishes' would also gain high likelihood values.

In fact, we validate this second point through a toy example. We provide an LLM with the question "X is cleaning dishes in the kitchen. What is X doing? a) washing plates, b) cleaning laundry, c) painting dishes. The correct choice is:" and calculate the likelihood for each of the 3 responses. The calculated likelihoods are 0.996, 0.006, 0.007 for a, b, c respectively (highest is selected), despite response (a) having no common words with the original statement unlike (b) and (c). This illustrates the ability of likelihood selection to capture semantic meanings.

### F.3 DETAILED PROMPTING EXAMPLE

We also note that while different choices are repeated along the batch, our likelihood implementation actually follows prior approaches where all answer candidates are fed together to the language model in addition to organizing the Q-A pairs in a batch dimension. Taking one simplified toy example, given a question "Where is the dog?" and answers "mat, table, bench", we use three queries along batch dimension as:

```
• Where is the dog?  Select the correct response from:  a)
  mat, b) table, c) bench.  The correct response is a) mat.

• Where is the dog?  Select the correct response from:  a)
  mat, b) table, c) bench.  The correct response is b) table.

• Where is the dog?  Select the correct response from:  a)
  mat, b) table, c) bench.  The correct response is c) bench.
```

In fact, applying likelihood selection without such prompting leads to significantly low performance for some datasets. We show this in Table 6 which we repeat here as Table A.5.

Table A.7: **LongVideoBench Evaluation:** We integrate MVU with the baseline from Abdin et al. (2024) and highlight the additional performance improvements achieved by our MVU framework.

| Method | Acc (%) |
|---|---|
| Phi-3-Vision-Instruct (Abdin et al., 2024) | 49.7 |
| Phi-3-Vision-Instruct + MVU | 50.4 |

Table A.8: **Ablation on Object Motion Trajectory (OMT) modality:** We perform this ablation on a different dataset given the motion focused aspect we explore. Accuracy (%) reported on the motion-based SSv2 dataset clearly indicate the usefulness of the OMT modality in our MVU framework.

| Method | OMT | Accuracy |
|---|---|---|
| Random | - | 0.6 |
| CLIP (Radford et al., 2021) | - | 4.0 |
| MAXI (Lin et al., 2023b) | - | 6.4 |
| MVU (ours) | ✗ | 3.6 |
| MVU (ours) | ✓ | **7.2** |

## G  OPEN-ENDED VIDEO QUESTION ANSWERING

In this section, we explore the ability of our proposed MVU framework to operate on open-ended video question answering (QnA) tasks. For this purpose, we evaluate on the Activity-Net dataset (Yu et al., 2019) reporting the accuracy metric. We follow evaluation settings identical to Maaz et al. (2023) for these evaluations.

Given the nature of open-ended QnA tasks (i.e. no answer choices, generate free form answers), we use standard generation instead of likelihood selection. We match the generated answers against ground-truth following (Maaz et al., 2023). We present these results in Table A.6 where our MVU achieves strong results and clear improvements over the similar LLM based approach from Zhang et al. (2023a). We compare against multiple recent approaches that use similar capacity LLMs VLMs for open-ended video QnA. We take these results as another indication to the generality of our MVU framework on video QnA tasks beyond MCQ style.

## H  LONGER VIDEO QUESTION ANSWERING

While established long video benchmarks used as the key evaluations in numerous prior work (Wang et al., 2025; 2024c; Min et al., 2024; Park et al., 2024; Zhang et al., 2023a; Kahatapitiya et al., 2024; Wang et al., 2023b) limit to roughly 1-3 minute long videos, some newer datasets include even longer videos (Wu et al., 2024). We explore such even longer videos by evaluating our method on the LongVideoBench dataset (Wu et al., 2024).

We select Phi-3-Vision-Instruct (Abdin et al., 2024) as our baseline since it is the best performing model we can replicate within our compute budget. We note that larger sized models using significantly larger context lengths are difficult to replicate within academic compute restraints. Results using this baseline from Abdin et al. (2024) and our MVU framework integrated over it are presented in Table A.7. MVU gains clear performance gains in this longer video dataset.

## I  ADDITIONAL ABLATIONS

In this section, we repeat part of our ablation from Table 5 focused on the object motion trajectory modality inputs. We note that common video QnA benchmarks require minimal understanding of object motion to answer most questions. Our goal is to explore the value of motion information in a more relevant tasks.

Therein we investigate a new motion focused dataset, Something-Something-v2 (Goyal et al., 2017) (SSv2), only for this single ablation. The SSv2 dataset focuses on motion-based category discrimination, providing an ideal evaluation to measure the usefulness of our object motion trajectory

Table A.9: **Frame Count Ablation:** We illustrate the importance of appropriate prompting when combining with likelihood selection, specifically for long video QnA tasks. Top-1 accuracy (%) is reported on EgoSchema subset (ES-S) and NextQA test set (NQA-T).

| Method | Frames | EgoSchema-S | Time (s) |
|--------|--------|-------------|----------|
| MVU | 16 | 60.3 | 2.42 |
| MVU | 32 | 60.4 | 2.48 |
| MVU | 64 | 60.4 | 2.60 |
| MVU | 128 | 61.2 | 2.81 |

Table A.10: **Context Length Comparison:** We compare the context length used (i.e. number of tokens) to achieve similar results with LLoVi (Zhang et al., 2023a) as opposed to our MVU. We achieve better performance utilizing less tokens.

| Method | Average Tokens | ES-F (%) |
|--------|----------------|----------|
| LLoVI | 1940 | 33.5 |
| MVU | 1124 | 37.6 |

modality. We benchmark on a subset of this dataset following (Lin et al., 2023b) and report these results in Table A.8. Our results while exceeding their performance also indicate the clear performance gains obtained when injecting the object motion trajectory modality into our MVU framework.

We also provide an ablation on frames used with our MVU framework in Table A.9. Increasing the number of frames leads to improved performance in contrast to some prior works (Mangalam et al., 2023) highlighting how our information fusion pipeline allows better utilization of the LLM context length. Additionally, the lightweight object detector and tracker used in MVU allows scaling the number of frames with a lesser increase in inference time.

## J  TOKENIZATION IN LLMS

Most modern LLMs utilize Byte-Pair Encoding (BPE) tokenization (Sennrich et al., 2015) to convert natural language into discrete tokens that are mapped to vocabulary embeddings. This process is learned from language itself and the resulting tokenization may sometimes break complete words into pieces (e.g. `example → ex-am-ple`). Given our utilization of logits extracted from an LLM forward pass, we note that each logit corresponds to a single token, which may at times be the embedding of some meaningless word piece. However, our calculation of a joint likelihood across a sequence of tokens ensures a meaningful process, which is validated by our strong results.

## K  LLM CONTEXT LENGTH

Using LLMs for long video understanding has proven successful (Zhang et al., 2023a; Wang et al., 2024c) but handling long context lengths remains a key issue (Mangalam et al., 2023; Park et al., 2024), often leading to lower performance when additional frame information is provided to the LLM. This draws importance to frame selection, but we argue that alternate forms of information bottlenecks can also provide improvements, often complementary to frame selection.

In MVU, instead of naively collecting all information within a frame, we only collect object centric spatial and motion information, allowing to process more frames at a fixed context length. In other words, MVU information extraction from multiple frames can be viewed as an alternative to frame selection. This is because our object information extraction indirectly acts as an information bottleneck similar to frame selection. For frames without objects of interest, no information is extracted. For multiple frames containing the same object (identified by our object tracker), the repetitive information is removed. This resembles the idea of selecting useful information from multiple frames.

In fact, when comparing the average token length for a similarly performing baseline (implemented under identical settings using a common LLM), we use less tokens (context length) to achieve similar results. We show these results in Table A.10.

Table A.11: **Multi-Frame LLaVA Baseline:** We implemented multi-frame variants of LLAVA (Liu et al., 2023a) with no video level training. Results indicate that without any video level training such naive extension does not lead to results improvements.

| Method | Frames | ES-S |
|--------|--------|------|
| LLaVA  | 1      | 55.8 |
| LLaVA  | 8      | 53.4 |
| LLaVA  | 16     | 46.2 |
| LLaVA  | 32     | 40.2 |

## L  ADDITIONAL BASELINES

We implement a multi-frame baseline directly using LLaVA-1.5 (Liu et al., 2023a) with no video level training. These results are reported in Table A.11. Results indicate that directly adding multiple frames to a VLM with no video level training does not lead to improved performance. Similar trends are observed in prior work (Kahatapitiya et al., 2024). These findings highlight the importance of careful per-frame information extraction and cross frame information fusion proposed in our MVU.

