# OpenReview forum: "Understanding Long Videos with Multimodal Language Models"
_ICLR.cc/2025/Conference — ICLR 2025 Poster_

### Official Review · Reviewer_NaUs · 2024-10-28

**Soundness:** 3
**Presentation:** 2
**Contribution:** 2
**Rating:** 5
**Confidence:** 4

**Summary:**

This paper proposes to inject video-specific information into an LLM based framework for video understanding. Specifically, the authors utilize off-the-shelf vision tools to extract three object-centric information modalities from videos and then leverage natural language as a medium to fuse the information. I think the proposed method seems novel and interesting. But I think this paper leaks a comparison with recent papers and the performance is not good enough.

**Strengths:**

The proposed method seems novel and I think that makes sense for video understanding. But I think the key frame selection plays the most important role for video understanding -- it seems the authors do not propose new method on this.

**Weaknesses:**

1.Figure 1 caption: “(left-right)” -> “(left-bottom)”;
2.I think the authors used both \cite{} and \citep{} in their writing;
3.Missing comparison with recent methods, for examples the publish papers in 2024. I think that is necessary to better your paper;
4.The performance seems not so good, only a marginly superiority compared to the counterparts even without comparing the recent methods. I think the performance is not good;
5.How to select the frames for most relevant frames? Is there any novelty to do that -- I think this is the most important part for video understanding.

**Questions:**

See the fifth point of weakneses.

---

> ### Author Response · Authors · 2024-11-23
> **Response to Reviewer NaUs**
>
> We thank the reviewer for the encouraging positive comments and all the suggestions to improve our work further. We address the points raised by the reviewer below.
>
> &nbsp;
>
> **W1,W2: “Typos”**
>
> Thank you for pointing these out. We will correct the Figure 1 caption and fix our \cite{} vs \citep{} usage.
>
> &nbsp;
>
>
> **W3: “Compare to 2024 papers”**
>
> Several recent works such as MoreVQA [1], VideoAgent [2], VideoTree [3], LifelongMemory [4], LangRepo [5], Tarsier [6], InternVideo2 [7], LVNet [8] achieve strong performance on long video understanding tasks by utilizing strong pre-trained LLM and MLM models. We will update Table 2 and Table 3 to include comparison to these works. We will also discuss these in our related work section.
> We would be highly grateful if the reviewer has pointers for any additional papers we may have missed.
>
> &nbsp;
>
>
> **W5: “Performance not good”**
>
> Our MVU is a framework that can be integrated with other works. We combine MVU with state-of-the-art LVNet (following their open-source implementation) to achieve additional improvements.
> | Method             | EgoSchema (full-set) |
> |--------------------|:---------:|
> | VideoTree          |    61.1   |
> | LifelongMemory     |    62.1   |
> | LangRepo           |    41.2   |
> | Tarsier            |    61.7   |
> | InternVideo2       |    60.2   |
> | MoreVQA            |    51.7   |
> | VideoAgent         |    54.1   |
> | LVNet              |    61.1   |
> | LVNet + MVU (ours) |    61.3   |
> |||
>
> This LVNet + MVU variant achieves strong performance competitive with more recent works. It also highlights how the MVU framework can be easily integrated with other existing works to achieve additional performance.
>
> &nbsp;
>
>
> **W5: “How to select the frames”**
>
> Frame selection is very important for long videos but somewhat orthogonal to our work. This means that we can integrate our MVU framework with existing frame-selection methods easily. LVNet is one such frame-selection approach. Our LVNet + MVU variant combines their orthogonal strengths to further improve performance.
> | Method             | EgoSchema |
> |--------------------|:---------:|
> | LVNet              |    61.1   |
> | LVNet + MVU (ours) |    61.3   |
> |||
>
> Secondly, MVU extracts object information from multiple frames. This can be viewed as an alternative in addition to frame selection.
> This is because our object information extraction indirectly acts as an information bottleneck similar to frame selection. For frames without objects of interest, no information is extracted. For multiple frames containing the same object (identified by our object tracker), the repetitive information is removed. This resembles the idea of selecting useful information from multiple frames.
> Within MVU, this process acts as a secondary form of information selection from many frames. We provide an ablation to highlight this below:
> | Method | Frames | EgoSchema-S | Time (s) |
> |--------|:------:|:-----------:|:--------:|
> | MVU    |   16   |     60.3    |   2.42   |
> | MVU    |   32   |     60.4    |   2.48   |
> | MVU    |   64   |     60.4    |   2.60   |
> | MVU    |   128  |     61.2    |   2.81   |
> |||||
>
> Note how our lightweight object detector and tracker allows us to increase the frames but maintain good inference speeds.
>
> &nbsp;
>
> Finally, we thank the reviewer again for these valuable comments that help improve our paper further. We are working on updating our manuscript with these changes.
>
> &nbsp;
> &nbsp;
>
> **References**
>
> [1] Min, Juhong, et al. "MoReVQA: Exploring Modular Reasoning Models for Video Question Answering." Proceedings of the IEEE/CVF Conference on Computer Vision and Pattern Recognition. 2024.
>
> [2] Wang, Xiaohan, et al. "Videoagent: Long-form video understanding with large language model as agent." European Conference on Computer Vision. Springer, Cham, 2024.
>
> [3] Wang, Ziyang, et al. "VideoTree: Adaptive Tree-based Video Representation for LLM Reasoning on Long Videos."
>
> [4]  Ying Wang, et al. "Lifelongmemory: Leveraging llms for answering queries in long-form egocentric videos."
>
> [5] Kahatapitiya, Kumara, et al. "Language repository for long video understanding.".
>
> [6] Wang, Jiawei, et al. "Tarsier: Recipes for training and evaluating large video description models.".
>
> [7] Wang, Yi, et al. "Internvideo2: Scaling video foundation models for multimodal video understanding.".
>
> [8] Park, Jongwoo et al. “Too Many Frames, not all Useful: Efficient Strategies for Long-Form Video QA.”.

---

> ### Comment · Reviewer_NaUs · 2024-11-26
> **Thanks for the responses**
>
> 1. Compared to LVNet, the proposed method only increases the baseline by 0.2, which is not convincing to verify the effectiveness. I even suspect that the performance might simply be attributed to random fluctuations.  2.  I believe this paper fails to address the core issues of using LLMs for video understanding, which makes its value fall below the acceptance threshold for ICLR.
> I choose to maintain my original score.

---

> ### Author Response · Authors · 2024-11-26
>
> We thank the reviewer for their response, but further emphasize the following 3 points about our work:
>
> &nbsp;
>
> **MVU improves on 20+ datasets**
>
> * MVU consistently achieves improved performance across 5 video QnA datasets (EgoSchema, NextQA, IntentQA, LongVideoBench, ActivityNet) as well as 16 robotics domain datasets.
> * While the performance gains are minimal in some cases, improvements are consistent across *20 different datasets* validating the effectiveness and generality.
> * LVNet is a state-of-the-art method for EgoSchema over which we obtain an additional improvement. Multiple very recent prior work like [1] and [2] from CVPR'24, ECCV'24 under perform this method by up to 10% points. We believe it is difficult for random fluctuations to improve performance over such a method. In contrast, MVU improves over LVNet even with a smaller margin and shows such improvement across multiple datasets.
>
> &nbsp;
>
> **Improvements over LVNet consistent**
>
> We further experiment with LVNet on NextQA and IntentQA datasets in addition to EgoSchema where MVU achieves consistent results improvements.
>
> | Method             | EgoSchema | NextQA | IntentQA |
> |--------------------|:---------:|:------:|:--------:|
> | LVNet              |    61.1   |  72.9  |   71.1   |
> | LVNet + MVU (ours) |    61.3   |  73.3  |   71.5   |
> |                    |           |        |          |
>
>
> This further validates how MVU is contributing to these improved results.
>
> &nbsp;
>
>
> **Core Issues of LLMs for video understanding**
>
> Our experiments and prior work suggest that core issues of LLMs are slow inference speed and handling long contexts.
> * MVU achieves faster inference for MCQ tasks with likelihood selection
> * MVU tackles long contexts with efficient information extraction
>
> The inference speed-up is highlighted in Tables 1, 6, and 7. The better handling of context length is highlighted in Table A.10 and Appendix K.
>
> &nbsp;
>
> We apologize if these points were not highlighted clearly earlier and would highly appreciate if the reviewer is able to take these into consideration. We are also updating the manuscript to better highlight these points.

---

> > ### Author Response · Authors · 2024-11-30
> > **Follow Up**
> >
> > Dear reviewer NaUs,
> >
> > We are highly thankful for the time and effort devoted by the reviewer to our paper and the following discussions. We are wondering if our last response addressed the concerns raised.
> >
> > Thanks you!

---

> ### Author Response · Authors · 2024-12-02
> **Follow Up 2**
>
> Dear reviewer NaUs,
>
> We are wondering if you were able to review our response to the concerns raised.
>
> Given the main concern being about results, we again highlight the following:
>
> &nbsp;
>
> **1) MVU improves over SOTA baseline LVNet across 3 datasets**
>
> Our `LVNet+MVU` shows improvements consistent across three datasets now. We also compare against two prior works to highlight how improving over this LVNet baseline is not easy.
>
> | Method             | EgoSchema | NextQA | IntentQA |
> |--------------------|:---------:|:------:|:--------:|
> MoreVQA (CVPR'24)          |    51.7   |  69.2  |   -   |
> VideoAgent (ECCV'24)       |    54.1   |  71.3  |   -   |
> | LVNet (NeurIPS'24)              |    61.1   |  72.9  |   71.1   |
> | LVNet + MVU (ours) |    **61.3**   |  **73.3**  |   **71.5**   |
> |                    |           |        |          |
>
> These results show the clear benefit of proposed MVU in multi-frame VLM settings with LVNet and how it outperforms SOTA prior works.
>
> &nbsp;
>
> **2) More multi-frame baselines on LongVideoBench**
>
> We integrate MVU with three different multi-frame baselines and evaluate on LongVideoBench dataset. MVU leads to performance improvements in each case.
>
> | Method         | Phi-3-Vision-Instruct | LLaVA-Next | VideoLLava |
> |----------------|:---------------------:|:---------------------:|:----------------------:|
> | Baseline       |          49.7         |    47.0    |    37.6    |
> | Baseline + MVU |          **50.4**         |    **47.8**    |    **39.2**    |
> |                |                       |            |            |
>
>
> &nbsp;
>
> We look forward to the reviewers response.
>
>
> Thank you!

---

### Official Review · Reviewer_a8nc · 2024-11-03

**Soundness:** 3
**Presentation:** 3
**Contribution:** 2
**Rating:** 6
**Confidence:** 3

**Summary:**

This paper proposes a training-free approach to understanding long-form videos by extracting explicit image-/object-level information. The extracted information is translated into natural language descriptions to make LLM ‘see’ the visual input. Experimental results on several videoQA benchmarks demonstrate the superiority.

**Strengths:**

1. The proposed likelihood selection approach offers a good way to speed up the inference in autoregressive LLMs.
2. Going beyond existing video datasets, this paper further evaluates the generalization ability on Open-X-Embodiedment.
3. The paper is well written and easy to follow.

**Weaknesses:**

1.	It seems that using likelihood as a selection criterion still focuses on the exact match between the generated text and the answer candidates in a per-logit manner, without considering the semantic meaning. For example, ‘C is washing plates’ vs ‘C is cleaning dishes’.
2.	Unlike prior approaches where all answer candidates are fed together to the language model, the proposed likelihood selection method organizes the Q-A pairs in a batch dimension. In this way, it seems that LLM fails to analyze the relationship between answer candidates, increasing the difficulty of QA.
3.	If the frames are uniformly sampled across the entire long video, how can you ensure the consistent occurrence of objects? In certain cases, the appeared objects in each frame are completely different. Another related question is whether using X_{OSL} and X_{OMT} extracted from densely sampled frames (i.e. with better object/trajectory consistency) would lead to performance gain.
4.	The authors are encouraged to evaluate the model on more long-video QA benchmarks, especially those designed to mitigate the language bias of existing long-video QA benchmarks (e.g., EgoSchema).
5.	In Tables 2 and 3, a series of recent state-of-the-art approaches are not compared. [1][2]

[1] Juhong Min et al. MoReVQA: Exploring Modular Reasoning Models for Video Question Answering, CVPR 2024.

[2] Xiaohan Wang et al. VideoAgent: Long-form Video Understanding with Large Language Model as Agent, ECCV 2024.

**Questions:**

Please refer to the weaknesses.

---

> ### Author Response · Authors · 2024-11-23
> **(1/2) Response to Reviewer a8nc**
>
> We thank the reviewer for the positive comments as well as all the valuable feedback. We address all concerns raised below.
>
> &nbsp;
>
> **W1: “Likelihood selection focuses on exact match”**
>
> Likelihood selection uses a likelihood measure which is the likelihood (probability) of the model generating the given sentence (as opposed to being an exact match). We apologize for any lack of clarity in our original paper regarding this and will highlight the following two points better in our paper.
>
> 1. In likelihood selection, we use the *likelihood measure* which is also used as the training loss when training LLMs. Given how LLMs trained with this loss (i.e. all decoder based LLMs such as LLaMA, Gemini, GPT) are highly effective at handling semantic meaning, it follows that this loss can capture semantic meaning.
>
> 2. This likelihood measure is calculated within the LLM latent space. This is equivalent to the probability (or likelihood) of that answer being generated by the LLM conditioned on the input question. We derive this in detail in Appendix F. Relating to the same example, this means that likelihood is an estimate of how likely the model would predict ‘C is washing plates’ as opposed to making that exact match. This means predictions closer to the target such as ‘C is cleaning dishes’ would also gain high likelihood values.
>
> In fact, we validate this second point through a toy example. We provide an LLM with the question *"X is cleaning dishes in the kitchen. What is X doing? a) washing plates, b) cleaning laundry, c) painting dishes. The correct choice is:"* and calculate the likelihood for each of the 3 responses. The calculated likelihoods are 0.996, 0.006, 0.007 for a, b, c respectively (highest is selected), despite response (a) having no common words with the original statement unlike (b) and (c). This illustrates the ability of likelihood selection to capture semantic meanings.
>
> We apologize for including less detail and lack of clarity regarding this in the main text and will update this more clearly in our final manuscript.
>
> &nbsp;
>
> **W2: “LLM fails to analyze the relationship between answer candidates”**
>
> Thank you for this interesting observation. Our likelihood implementation actually follows prior approaches where all answer candidates are fed together to the language model in addition to organizing the Q-A pairs in a batch dimension. This is illustrated in Table A.4 - we will provide more examples to highlight this better during revision.
> Taking one simplified toy example, given a question “Where is the dog?” and answers “mat, table, bench”, we use three queries along batch dimension as:
> * Where is the dog? Select the correct response from: a) mat, b) table, c) bench. The correct response is a) mat
> * Where is the dog? Select the correct response from: a) mat, b) table, c) bench. The correct response is b) table
> * Where is the dog? Select the correct response from: a) mat, b) table, c) bench. The correct response is c) bench
>
> In fact, applying likelihood selection without such prompting leads to significantly low performance for some datasets. We show this in Table 6 which we repeat below:
> | Dataset                          | ES-S | NQA-T |
> |----------------------------------|:----:|:-----:|
> | No answer candidates in prompt   | 58.2 |  35.8 |
> | With answer candidates in prompt | 60.3 |  55.4 |
> ||||
>
> We apologize for any lack of clarity in our paper and will update to highlight this better.
>
> &nbsp;
>
> **W3a: “Ensure consistent occurrence of objects”**
>
> This is an important direction for long video understanding. However, our work is orthogonal to this direction. In fact, taking prior work like LVNet which directly addresses such issues, we can integrate our MVU framework to further improve their performance. We include these results below.
> | Method             | EgoSchema (full set) |
> |--------------------|:---------:|
> | LVNet              |    61.1   |
> | LVNet + MVU (ours) |    61.3   |
> |||
>
> &nbsp;
>
> **W3b: “Performance on more densely sampled frames”**
>
> We run the following ablations focussed on densely sampled frames. Dense sampling improves performance.
> | Method | Frames | EgoSchema-S | Time (s) |
> |--------|:------:|:-----------:|:--------:|
> | MVU    |   16   |     60.3    |   2.42   |
> | MVU    |   32   |     60.4    |   2.48   |
> | MVU    |   64   |     60.4    |   2.60   |
> | MVU    |   128  |     61.2    |   2.81   |
> |||||
>
> In fact, the lightweight object detector and tracker used in MVU allows scaling the number of frames with a lesser increase in inference time.

---

> ### Author Response · Authors · 2024-11-23
> **(2/2) Response to Reviewer a8nc**
>
> **W4: “Results on Datasets better than EgoSchema”**
>
> We evaluate on the LongVideoBench dataset and provide these results below. We select this `Phi-3-Vision-Instruct` baseline since it is the best performing model we can replicate within our compute budget.
>
> | Method                      | LongVideoBench |
> |-----------------------------|:--------------:|
> | Phi-3-Vision-Instruct       |      49.7      |
> | Phi-3-Vision-Instruct + MVU |      50.4      |
> |||
>
> It is clear how MVU combined with this baseline can improve performance.
>
> &nbsp;
>
> **W5: “Recent state-of-the-art approaches”**
>
> We thank the reviewer for pointing these works to us. We are updating Tables 2 and 3 to include these comparisons. We also repeat some results below for quick reference.
>
> | Method             | EgoSchema (full-set) |
> |--------------------|:---------:|
> | MoreVQA [1]        |    51.7   |
> | VideoAgent [2]     |    54.1   |
> | VideoTree [3]        |    61.1   |
> | LifelongMemory [4]     |    62.1   |
> | LangRepo [5]       |    41.2   |
> | Tarsier [6]             |    61.7   |
> | InternVideo2 [7]   |    60.2   |
> | LVNet [8]              |    61.1   |
> | LVNet + MVU (ours) |    61.3   |
> |||
>
> Here we apply ours over LVNet given its state-of-the-art performance, open-source implementation, and publicly available pre-computed captions (computing long video frame captions with closed source models can be expensive).
>
> &nbsp;
>
> We thank the reviewer again for all valuable feedback helpful for improving our paper further.
>
> &nbsp;
> &nbsp;
>
> **References**
>
> [1] Min, Juhong, et al. "MoReVQA: Exploring Modular Reasoning Models for Video Question Answering." Proceedings of the IEEE/CVF Conference on Computer Vision and Pattern Recognition. 2024.
>
> [2] Wang, Xiaohan, et al. "Videoagent: Long-form video understanding with large language model as agent." European Conference on Computer Vision. Springer, Cham, 2024.
>
> [3] Wang, Ziyang, et al. "VideoTree: Adaptive Tree-based Video Representation for LLM Reasoning on Long Videos."
>
> [4]  Ying Wang, et al. "Lifelongmemory: Leveraging llms for answering queries in long-form egocentric videos."
>
> [5] Kahatapitiya, Kumara, et al. "Language repository for long video understanding.".
>
> [6] Wang, Jiawei, et al. "Tarsier: Recipes for training and evaluating large video description models.".
>
> [7] Wang, Yi, et al. "Internvideo2: Scaling video foundation models for multimodal video understanding.".
>
> [8] Park, Jongwoo et al. “Too Many Frames, not all Useful: Efficient Strategies for Long-Form Video QA.”.

---

> > ### Comment · Reviewer_a8nc · 2024-11-25
> > **Official Comment by Reviewer a8nc**
> >
> > With these detailed clarifications, most of my concerns have been solved. I will consider increase my score

---

> > > ### Author Response · Authors · 2024-11-26
> > >
> > > We thank the reviewer for their positive comments and appreciate the positive rating to our paper.

---

> > > > ### Author Response · Authors · 2024-11-27
> > > > **Updated Manuscript**
> > > >
> > > > We thank Reviewer a8nc once again for the valuable feedback and positive comments. We have now updated the PDF with all proposed changes in the comments and addressed all concerns raised by the reviewer.
> > > >
> > > > While we thank the reviewer for the positive rating, we inquire if there are any remaining concerns regarding the paper that we could address. We would be highly grateful for any additional feedback that could further strengthen our paper and improve its score.
> > > >
> > > > Thank you again for your time and effort devoted towards reviewing our paper.

---

> > > > > ### Author Response · Authors · 2024-11-30
> > > > > **Additional Results**
> > > > >
> > > > > We thank the reviewer once again for all their feedback and support in improving our paper. We present details for two more sets of experiments (using multi-frame baselines) to further highlight the generality of MVU framework.
> > > > >
> > > > > &nbsp;
> > > > >
> > > > > **LVNet + Ours:**
> > > > >
> > > > > We integrate MVU with LVNet baseline and evaluate on 3 benchmarks to establish performance improvements. MVU achieves improvements consistent across all three datasets. We also include two recent prior works for comparison purposes.
> > > > >
> > > > > | Method             | EgoSchema | NextQA | IntentQA |
> > > > > |--------------------|:---------:|:------:|:--------:|
> > > > > MoreVQA          |    51.7   |  69.2  |   -   |
> > > > > VideoAgent       |    54.1   |  71.3  |   -   |
> > > > > | LVNet              |    61.1   |  72.9  |   71.1   |
> > > > > | LVNet + MVU (ours) |    **61.3**   |  **73.3**  |   **71.5**   |
> > > > > |                    |           |        |          |
> > > > >
> > > > > These results show the clear benefit of proposed MVU in multi-frame VLM settings.
> > > > >
> > > > > &nbsp;
> > > > >
> > > > > **LongVideoBench Evaluation:**
> > > > >
> > > > > We also run more evaluations with MVU on the LongVideoBench dataset that is designed mitigate language biases in video QnA. We integrate MVU over three different baselines and show consistent performance improvements in each case.
> > > > >
> > > > > | Method         | Phi-3-Vision-Instruct | LLaVA-Next | VideoLLava |
> > > > > |----------------|:---------------------:|:---------------------:|:----------------------:|
> > > > > | Baseline       |          49.7         |    47.0    |    37.6    |
> > > > > | Baseline + MVU |          **50.4**         |    **47.8**    |    **39.2**    |
> > > > > |                |                       |            |            |
> > > > >
> > > > > &nbsp;
> > > > >
> > > > > We first thank the reviewer again for the suggestions to explore such directions. We hope these additional results help resolve any remaining concerns regarding our paper.

---

### Official Review · Reviewer_V8Po · 2024-11-05

**Soundness:** 3
**Presentation:** 3
**Contribution:** 2
**Rating:** 6
**Confidence:** 4

**Summary:**

This paper introduces two baselines and a novel approach called the Multimodal Video Understanding (MVU) framework for video understanding tasks. The baselines explore using either a single frame or no visual input at all. In contrast, MVU aggregates multimodal information relevant to the video to enhance understanding.

**Strengths:**

+ It is interesting that this work introduces three key attributes essential for video understanding: Global Object Information, Object Spatial Location, and Object Motion Trajectory. These attributes contribute significantly to a more comprehensive analysis of video content.

**Weaknesses:**

- Although this paper focuses on long-video understanding, it lacks specific design elements to address long-video scenarios. Challenges such as context length limiting input frames and modeling long-range temporal information are not directly addressed. The attributes introduced—Global Object Information (GOI), Object Spatial Location (OSL), and Object Motion Trajectory (MOT)—are not tailored to tackle these issues in long-video understanding.

- In Table 1, the comparisons may be weaker due to the differing training setups and base models used across experiments. As such, the superior performance of SF-LLM over the state-of-the-art does not necessarily imply that the number of frames is irrelevant for video understanding.

- Since the model is based on LLaVA-v1.5-13B, an important baseline is missing: the use of multiple frames as input for LLaVA-v1.5-13B.

- Certain details are lacking, such as the method for using LLaVA-v1.5-13B for frame sampling, as mentioned in Figure 3.

- Likelihood Selection is widely used in the MCQ benchmark as an additional track. For a fairer comparison, other methods should also incorporate this strategy for comparison results.

- The benchmark in this paper only includes mid-length videos, roughly under three minutes. To more competitively demonstrate MVU's capabilities in long-video understanding, it would be beneficial to evaluate on benchmarks like VideoMME and LongVideoBench.

**Questions:**

As mentioned in the weakness.

---

> ### Author Response · Authors · 2024-11-23
> **(1/2) Response to Reviewer V8Po**
>
> We thank the reviewer for their feedback and address all concerns below.
>
> &nbsp;
>
> **W1a: “Context length limiting input frames not addressed”**
>
> It is unclear what context length here means. Assuming it is about LLM context lengths (since we use an LLM in our work), we reiterate that this is specifically a key issue addressed in our work. Instead of naively collecting all information within a frame, our MVU only collects object centric spatial and motion information, allowing to process more frames at a fixed context length. As described in our method and empirically validated in our ablations (see L518-521) our work directly addressed this issue.
>
> In fact, when comparing the average token length for a similarly performing baseline (implemented under identical settings using a common LLM), we use less tokens (context length) to achieve similar results.
> | Method | Average Tokens | ES-F (%) |
> |--------|----------------|----------|
> | LLoVI  | 1940           | 33.5     |
> | MVU    | 1124           | 37.6     |
> ||||
>
> This highlights how MVU acts as an information bottleneck to extract useful information - a key design element to support long video processing.
>
> &nbsp;
>
> **W1b: “Modeling long-range temporal information not addressed”**
>
> We rely on the power of the LLM to capture temporal information. We provide our video as a sequence of tokens so the temporal information can be captured by the LLM. This is motivated by prior work such as LLoVi. Fusing our three information types and providing to the LLM such that it can perform all temporal modeling is another key design element enabling long video understanding.
>
> Applying additional techniques for more long-range modeling is orthogonal to our work. In fact, we apply MVU over one such work, LVNet [1] that is capable of processing up to 900 frames. Their extracted information is combined with MVU information to make final predictions, leading to additional performance improvements as shown:
>
> | Method             | EgoSchema |
> |--------------------|:---------:|
> | LVNet [1]          |    61.1   |
> | LVNet + MVU (ours) |    61.3   |
> |||
>
> &nbsp;
>
> **W4: “Table 1 comparisons may be weaker due to the differing training setups and base models”**
>
> All numbers in Table 1 are from experiments that we run using identical (pre-trained, publicly available) base models and evaluation settings. There is no training of the pre-trained models in our framework or the baselines used in Table 1. Our reproduced numbers for baselines are also comparable with those in other works [2], validating our implementations. We have already given careful attention to ensure no such discrepancies occur.
>
> &nbsp;
>
> **W5: “Multi-Frame LLaVa Baseline”**
>
> Directly adding more frames to the LLaVA baseline we use (that is trained only on images) does not result in improvements for our tasks.
>
> | Method | Frames | ES-S |
> |--------|:------:|:----:|
> | LLaVA  |    1   | 55.8 |
> | LLaVA  |    8   | 53.4 |
> | LLaVA  |   16   | 46.2 |
> | LLaVA  |   32   | 40.2 |
> ||||
>
> This reduction of performance by naively adding more information (as frames) is consistent with observations in prior work [3].
>
> &nbsp;
>
> **W6: “Frame Selection Details”**
> As mentioned in L248, L297, L916, and L955 that we perform uniform frame sampling. We apologize for any clarity issues in Figure 3 and will revise this as shown above.

---

> > ### Author Response · Authors · 2024-11-23
> > **(2/2) Response to Reviewer V8Po**
> >
> > **W7: “Likelihood Selection is widely used in the MCQ benchmark as an additional track”**
> >
> > Adapting likelihood selection to long video is part of our contribution and we show that it leads to better performance in our ablations.
> > Although this is used widely in NLP, it is much less used in images and we are not aware of any video based works using this.
> > We respectfully request the reviewer to provide any references to prior video QnA work that utilizes likelihood selection in any capacity.
> >
> > &nbsp;
> >
> > **W8: “Only three minute long videos”**
> >
> > We focus on established long video benchmarks used as the key evaluations in numerous prior work [1,2,4,5,6,7,8,9].
> > Video QnA traditionally involves short (10-20 second) clips (see Maaz et al. (2023)) - we evaluate on such datasets as well (see Table A.5). Our focus in this work is on longer videos averaging 1-2 minutes, while also showing results on the traditional 10-20 second long clips.
> >
> > Nevertheless, we explore even longer videos by evaluating our method on LongVideoBench. Results are provided below:
> >
> > | Method                      | LongVideoBench |
> > |-----------------------------|:--------------:|
> > | Phi-3-Vision-Instruct       |      49.7      |
> > | Phi-3-Vision-Instruct + MVU |      50.4      |
> > |||
> >
> >
> > We select `Phi-3-Vision-Instruct` as the baseline since it is the best performing model we can replicate within our compute budget. Larger sized models using significantly larger context lengths are difficult to replicate within academic compute restraints.
> >
> > &nbsp;
> > &nbsp;
> >
> > **References**
> >
> > [1] Park, Jongwoo et al. “Too Many Frames, not all Useful: Efficient Strategies for Long-Form Video QA.”
> >
> > [2] Kahatapitiya, Kumara, et al. "Language repository for long video understanding."
> >
> > [3] Mangalam, Karttikeya, Raiymbek Akshulakov, and Jitendra Malik. "Egoschema: A diagnostic benchmark for very long-form video language understanding." Advances in Neural Information Processing Systems 36 (2023).
> >
> > [4] Min, Juhong, et al. "MoReVQA: Exploring Modular Reasoning Models for Video Question Answering." Proceedings of the IEEE/CVF Conference on Computer Vision and Pattern Recognition. 2024.
> >
> > [5] Wang, Xiaohan, et al. "Videoagent: Long-form video understanding with large language model as agent." European Conference on Computer Vision. Springer, Cham, 2024.
> >
> > [6] Wang, Ziyang, et al. "VideoTree: Adaptive Tree-based Video Representation for LLM Reasoning on Long Videos."
> >
> > [7] Ying Wang, et al. "Lifelongmemory: Leveraging llms for answering queries in long-form egocentric videos."
> >
> > [8] Wang, Jiawei, et al. "Tarsier: Recipes for training and evaluating large video description models.".
> >
> > [9] Wang, Yi, et al. "Internvideo2: Scaling video foundation models for multimodal video understanding.".

---

> > > ### Author Response · Authors · 2024-11-27
> > > **Follow-up on Reviewer V8Po**
> > >
> > > Dear Reviewer V8Po,
> > >
> > > Thank you again for your feedback and time/effort reviewing our paper. Since the rebuttal period is ending soon, please let us know if our responses have addressed your concerns. We are happy to engage in further discussion to provide more clarifications if needed.
> > >
> > > Thank you!

---

> > > > ### Comment · Reviewer_V8Po · 2024-11-27
> > > >
> > > > Thank you for your thoughtful response, which solves most of my concerns.
> > > >
> > > > I appreciate the contribution of this paper, particularly its identification of three key types of information—Global Object Information, Object Spatial Location, and Object Motion Trajectory—that enhance zero-shot video understanding. These insights are valuable for improving single-frame models in video understanding.
> > > >
> > > > However, when comparing LVNet + MVU with LVNet in the multi-frame setting (72.9 vs. 73.3), the impact of these contributions appears less pronounced, suggesting that the benefits may be more limited for multi-frame models.
> > > >
> > > > That said, I acknowledge the overall significance of this paper's contributions and have decided to increase my score accordingly.

---

> > > > > ### Author Response · Authors · 2024-11-27
> > > > >
> > > > > We thank the reviewer again for their time & efforts as well as all the valuable feedback that further strengthened our paper. We are also grateful for the positive rating to our paper.

---

> > > > > > ### Author Response · Authors · 2024-11-30
> > > > > > **Additional Results**
> > > > > >
> > > > > > We thank the reviewer again for all their feedback and the positive rating to our paper. To address the remaining concern regarding the `LVNet+MVU` setup, we run some more additional experiments using several multi-frame baselines.
> > > > > >
> > > > > > &nbsp;
> > > > > >
> > > > > > **IntentQA results with LVNet**
> > > > > >
> > > > > > We present results on the IntentQA dataset as well. Our `LVNet+MVU` shows improvements consistent across three datasets now. We also compare against two prior works to highlight how improving over this LVNet baseline is not easy.
> > > > > >
> > > > > > | Method             | EgoSchema | NextQA | IntentQA |
> > > > > > |--------------------|:---------:|:------:|:--------:|
> > > > > > MoreVQA          |    51.7   |  69.2  |   -   |
> > > > > > VideoAgent       |    54.1   |  71.3  |   -   |
> > > > > > | LVNet              |    61.1   |  72.9  |   71.1   |
> > > > > > | LVNet + MVU (ours) |    **61.3**   |  **73.3**  |   **71.5**   |
> > > > > > |                    |           |        |          |
> > > > > >
> > > > > > These results show the clear benefit of proposed MVU in multi-frame VLM settings with LVNet.
> > > > > >
> > > > > > &nbsp;
> > > > > >
> > > > > > **More multi-frame baselines on LongVideoBench**
> > > > > >
> > > > > > We integrate MVU with three different multi-frame baselines and evaluate on LongVideoBench dataset. MVU leads to performance improvements in each case.
> > > > > >
> > > > > > | Method         | Phi-3-Vision-Instruct | LLaVA-Next | VideoLLava |
> > > > > > |----------------|:---------------------:|:---------------------:|:----------------------:|
> > > > > > | Baseline       |          49.7         |    47.0    |    37.6    |
> > > > > > | Baseline + MVU |          **50.4**         |    **47.8**    |    **39.2**    |
> > > > > > |                |                       |            |            |
> > > > > >
> > > > > >
> > > > > > &nbsp;
> > > > > >
> > > > > > We thank the reviewer again for their valuable feedback and all time devoted towards reviewing and follow-up discussion of our paper. We hope these additional evaluations help resolve any remaining concerns about our paper.

---

### Author Response · Authors · 2024-11-23
**General Rebuttal**

We first thank all reviewers for the positive feedback:
*Likelihood selection approach offers a good way to speed up the inference. Paper further evaluates the generalization ability* (**R-a8nc**).
*Proposed method seems novel and I think that makes sense for video understanding* (**R-NaUs**).

Next we reiterate our key contributions as,
1. Highlighting performance of modality constrained baselines
2. Adapting likelihood selection to long video tasks
3. Video QnA framework MVU

We highlight how the proposed MVU is a framework that can be combined with existing approaches. Our main comparison was to LLoVi, where integrating MVU achieves both accuracy and inference speed improvements. These results cover multiple datasets including in the robotics domain establishing generality of MVU.

In our rebuttal, we provide evaluations for MVU combined with a more recent approach (LVNet) that involves explicit frame selection and achieves much stronger results. The combined LVNet+MVU variant further improves performance. This highlights how MVU can be integrated with existing methods for additional improvement.

We also highlight how MVU, especially with more dense sampling of frames, implicitly acts as an information bottleneck similar to frame selection works.

We provide more details on these as we respond to each reviewer. We hope the rebuttal is able to better establish the contributions of our paper and how it can be useful to the computer vision community.

---

> ### Author Response · Authors · 2024-11-27
> **Updated Manuscript**
>
> **Updated Manuscript:** We have uploaded a new PDF including changes (colored in green) to address all concerns raised by reviewers. Key changes include:
>
> 1. Added newer baselines from 2024 and integrating MVU over those baselines (updates to `Related Work` & `Experiments` sections, `Table 2`, `Table 3`)
> 2. Clarify nature and role of Likehood Selection (updates to `Sec 3.2`, `Appendix F.2, F.3`)
> 3. Evaluations on datasets containing very long ( > 3 mins) videos (`Appendix H`)
> 4. Detailed discussion on LLM context length (`Appendix K`)
> 5. Additional ablations and baselines (`Appendix I, L`)
>
> &nbsp;
>
> **NOTE:**
> Several of the newly added papers (from 2024) are concurrent work (i.e. published after July 1st, the date set in [ICLR guidelines](https://iclr.cc/Conferences/2025/ReviewerGuide#:~:text=A%3A%20We%20consider%20papers%20contemporaneous,own%20work%20to%20that%20paper.)). In fact, some of these works even cite and include results from our paper.

---

### Meta-Review · Area_Chair_HiAV · 2024-12-20

**Metareview:**

This paper introduces the Multimodal Video Understanding (MVU) framework, which leverages object-centric information (Global Object Information, Object Spatial Location, and Object Motion Trajectory) to enhance video understanding by translating visual data into natural language for use in large language models (LLMs). While the novel approach demonstrates marginal improvements in mid-length video QA benchmarks, reviewers highlight several limitations. Key concerns include the lack of focus on long-video-specific challenges, such as modeling long-range temporal dependencies, and the absence of comparisons with recent state-of-the-art methods. Additionally, the methodology lacks innovation in critical areas, such as frame selection, and the performance gains are only marginal. The paper is generally well-written but would benefit from evaluations on more competitive long-video QA benchmarks and more substantial comparisons to recent advancements. Overall, the work is promising but falls short in execution and impact.

**Additional Comments On Reviewer Discussion:**

The reviewers acknowledge the paper's contribution in identifying three key types of information—Global Object Information, Object Spatial Location, and Object Motion Trajectory—that enhance zero-shot video understanding, particularly for single-frame models. However, concerns remain about the limited performance improvement (e.g., a marginal gain of 0.2 when comparing LVNet + MVU to LVNet) and the lack of convincing evidence for its effectiveness in multi-frame settings. One reviewer appreciates the insights and increases their score, while another criticizes the paper for failing to address core challenges in applying LLMs to video understanding, maintaining a negative stance. The contributions are noted, but the limited impact and unresolved issues keep opinions mixed. The meta-reviewer read the authors' feedback carefully and found the paper still deserves exposure.

---

### Decision · Program_Chairs · 2025-01-22

Accept (Poster)